# Behavioural compatibility, not fear, best predicts the looking patterns of chacma baboons
Andrew T. L. Allan [1,2] ✉, Laura R. LaBarge[2,3], Annie L. Bailey[2], Benjamin Jones[2], Zachary Mason[2], Thomas Pinfield[2], Felix Schröder[2], Alex Whitaker[2], Amy F. White[1,2], Henry Wilkinson [2] & Russell A. Hill[1,2,4]

Animal vigilance is often investigated under a narrow set of scenarios, but this approach may overestimate its contribution to animal lives. A solution may be to sample all looking behaviours and investigate numerous competing hypotheses in a single analysis. In this study, using a wild group of habituated chacma baboons (*Papio ursinus griseipes*) as a model system, we implemented a framework for predicting the key drivers of looking by comparing the strength of a full array of biological hypotheses. This included methods for defining individual-specific social threat environments, quantifying individual tolerance to human observers, and incorporating predator resource selection functions. Although we found evidence supporting reactionary and within-group (social) vigilance hypotheses, risk factors did not predict looking with the greatest precision, suggesting vigilance was not a major component of the animals' behavioural patterns generally. Instead, whilst some behaviours constrain opportunities for looking, many shared compatibility with looking, alleviating the pressure to be pre-emptively vigilant for threats. Exploring looking patterns in a thorough multi-hypothesis framework should be feasible across a range of taxa, offering new insights into animal behaviour that could alter our concepts of fear ecology.

Vigilance, visually monitoring surroundings for possible dangers and difficulties, is a behaviour used by many animals to avoid costly interactions with threats[1]. Vigilance has been investigated in a wide range of taxa, but as research has grown, so too has the diversity of definitions used to sample vigilance[1,2]. Concomitantly, analytical approaches have also diversified, which combined with definitional variation, make cross-study comparisons challenging[2]. Despite animal vigilance lacking a clear unified research framework, a common approach has been to sample 'vigilance' directly, often under a narrow set of scenarios, and to explore a small number of hypotheses in isolation[1,2]. This approach is problematic as it does not help disentangle the numerous competing hypotheses and may thus overestimate the extent to which fear regulates animal lives. A preferable approach is to employ a framework that explores numerous factors in unison, allowing researchers to gain a more intricate understanding of the relative weighting each factor has and their contribution to behavioural patterns generally[3]. So far, this approach has been underutilised, particularly in vigilance research[2].

There are now numerous context-dependent and often competing vigilance hypotheses for researchers to consider[1,2]. For example, it has long been viewed that vigilance limits how much time an animal can spend engaged in other fitness-enhancing activities such as foraging[4]. This notion underpins the group-size effect on vigilance[5], whereby group-living animals are hypothesized to circumnavigate the foraging-vigilance trade-off by diminishing their individual investment in vigilance as group-size increases. Yet, vigilance may also increase as group-size increases[6], owing to greater within-group competition and conflict. For example, dominance rank can influence vigilance patterns exhibited by group members in complex social systems[7]. Group cohesion and individual spatial position can then enhance or diminish dilution[8,9] and confusion[10] effects, whilst food intake rate and food availability can interact with risk hypotheses - as animals foraging in areas with more food are expected to decrease investment in vigilance to maximise energy intake[11]. These predictions may also change based on the sensory capacity an animal has during different behaviours, as some species can use their peripheral vision during foraging to detect or monitor threats[12]. Similarly, many species have the sensory capacity to detect localised threats during engaged behaviours such as foraging, despite not being overtly vigilant[13,14]. Conversely, foraging

[1]Department of Anthropology, Durham University, Durham, UK. [2]Primate and Predator Project, Lajuma Research Centre, Louis Trichardt, South Africa. [3]Department for the Ecology of Animal Societies, Max Planck Institute of Animal Behavior, Konstanz, Germany. [4]Department of Zoology, University of Venda, Thohoyandou, South Africa. ✉e-mail: andrewtlallan@hotmail.com

tasks requiring increased attention and handling time can hinder threat detection substantially[13]. Vigilance predictions are therefore inextricably linked to a specific study species' postures and sensory capacity, and the specific foraging tasks they encounter.

Given these complex, interacting, and context-dependent predictions, several authors have attempted to tease apart different forms of vigilance and sample them directly, including social/non-social vigilance[15], social/antipredator[16], pre-emptive/reactionary[17], and induced/routine[18]. The clear drawback of this approach is that the multitude of vigilance hypotheses may warrant a potentially endless list of subtypes of vigilance to be defined. Additionally, under field conditions, it is very challenging to precisely and consistently identify when an animal is performing specific vigilance behaviours (e.g., pre-emptive or social vigilance)[2,19]. This is largely because few animals flawlessly betray an internal state of vigilance or their precise focus of visual attention on a consistent basis[1,20].

Given that different definitions can also vary in their likelihood of achieving inter-rater agreement[19], increasing the breadth and specificity of vigilance definitions will likely introduce error into findings whilst also making cross-study comparisons very challenging[2]. Allan & Hill[2,19] suggested that researchers consider adopting a single definition and research approach, presenting the looking definition and framework as an option. In this approach, observers record all looking behaviours across a full range of scenarios, regardless of the study animals' posture (e.g., head up) or internal state (e.g., vigilant, cautious etc). The contextual information from each observation can then be used to identify the most prominent trends analytically, allowing researchers to explore the relative weighting of a full array of hypotheses in a single consolidated analysis[2]. Looking has been associated with greater inter-observer reliability than other vigilance-specific definitions[19], but the research framework has yet to be fully implemented.

In this study, using a wild group of habituated chacma baboons (*Papio ursinus griseipes*) as a model system, we implemented the looking framework to disentangle the various subcomponents of vigilance (e.g., within-group vigilance, pre-emptive vigilance for predators, observer vigilance) from non-risk driven looking patterns (e.g., behaviour, compatible handling/feeding time) and weight them according to their relative prediction accuracy. This included new methods for quantifying the local social threat environment and observer tolerance for each individual and incorporating spatial variables for predator habitat use. We found little evidence supporting pre-emptive risk hypotheses (e.g., looking was not elevated in areas where predation risk was high), but did find evidence supporting reactionary (e.g., encountering another group) and within-group (social) vigilance hypotheses; however, these risk factors predicted looking with less precision than foraging (e.g., feeding rate) and compatibility (e.g., specific foraging task) factors. In particular, certain foraging items/tasks (e.g., biting seeds or manipulating roots with their hands) offered moments of compatible-looking time, which the baboons readily used, regardless of the ecological scenario. As baboons have the capacity to collect multiple types of information concurrently (e.g., detecting an approaching threat despite looking at another stimuli)[14], there is unlikely to be a consistent need for them to adjust their looking patterns according to pre-emptive risk scenarios. Together, our approach suggests that studying looking in a thorough multi-hypothesis framework can advance our understanding of the role fear plays in regulating animal lives.

## Results

### Research framework
We initially employed an information-theoretic approach[21] to identify these competing hypotheses and then created a set of independent models to represent each of the key biological hypotheses for looking (see Table 1 and Supplementary Table S1 and Text S1 for detailed justifications). No single risk model included more than one type of risk variable, allowing insights into whether certain patterns of behaviour (e.g., time spent engaged/not engaged, spatial position) can independently produce different influences on looking patterns depending on the risk type.

### Stacking weights and Bayesian $R^2$ estimates
We found the greatest prediction accuracy for looking frequency in models incorporating data on an animal's feeding rate/food item, specific behaviours, habitat type (e.g., forest, woodland, grassland etc), home-range familiarity (core, frequently used, and boundary areas), and the number of within-group threats (within 5 meters), although several other models shared lower weight (Table 2). When the models with at least 0.001 weight were re-stacked, those including specific behaviours and feeding rate/items shared 0.863 of the model weights and were the only models to produce $R^2$ estimates greater than 0.1 - suggesting that these factors were the most accurate for predicting the frequency of looking. Moderate initial weights (and $R^2$ values close to zero) for models exploring within-group threats, habitat type, and home-range familiarity suggests these models likely predicted some points with high precision but produced less accurate predictive distributions of looking frequency overall. The remaining factors are unlikely to be consistent drivers of looking frequency as they did not consistently yield greater weight than the intercept, minimal, or group geometry and cohesion models (see Supplementary Tables S2–S7 for model summaries).

The $R^2$ estimates for duration models were all greater than 0.5 (excluding the intercept-only model) and their initial stacking weights indicated that specific behaviours, feeding rate/items, time since male calls associated with threats (wahoos), time since extra-group/within-species encounters (e.g., another group or a lone foreign individual), and within-group threats held the greatest prediction accuracy for the total duration of looking. When models with at least 0.001 weight were re-stacked, the models exploring specific behaviours, feeding rate/items, time since male vocalisations, and time since extra-group/within-species encounters shared 0.899 of the model weights. These models were therefore considered to be the most accurate and consistent predictors of looking duration, although within-group threats may also be important considering its initial weighting.

We observed a small difference between conditional and marginal $R^2$ estimates, suggesting that the group-level structure (i.e., observation date crossed with individual identity) did not significantly improve the predictive performance of our models. In several cases, models exhibited less weight after the stacks were simplified. This is because of the stacking procedure, whereby similarly performing models (that share some of the same predictors) have their weight combined to the model exhibiting greater predictive accuracy[22]. For example, for the frequency response variable, the stacking procedure likely combined the weights for the habitat type model (model 17) with other similar models using some of the same predictors (e.g., models 9, 10, and 16). This indicates that the habitat model produces a predictive distribution with greater accuracy than these similar models but still has far lower prediction accuracy than specific behaviours and feeding rate/items models.

### Feeding rate and food items (model 4)
We found a positive association between feeding rate (number of bites or items consumed) and the frequency of looking and a negative relationship between feeding rate and duration of looking (see Fig. 1 and Supplementary Tables S8 and S9 for model summaries). The main food item consumed, foraged, or manipulated during the observation was also important for both variables - certain species of seeds were associated with lower durations but more frequent bouts of looking, whilst feeding on leaves and grass blades/seeds was associated with longer durations (Fig. 2). Collectively these results suggest that foraging tasks and their relative complexity, success, and compatibility with looking are key factors governing behavioural patterns; the positive association between frequency of looking and feeding rate reiterates that some foraging tasks can promote looking.

### Specific behaviours (model 5)
The frequency of looking was positively associated with biting, handling, picking, scratching, and movement, but negatively associated with grooming another animal. All other behaviours produced credible intervals

**Article**

**Table 1 | Research framework for investigating the looking patterns of a habituated group of chacma baboons**

| Model | Population-level effects | Model purpose/hypothesis |
|---|---|---|
| 1 | ~1 | Intercept-only model |
| 2 | Age-sex class + Behaviour | Minimal model |
| **Group geometry and cohesion** | | |
| 3 | Number of neighbours within 5 meters * Spatial position + Age-sex class + Behaviour | A: Pre-emptive vigilance when isolated/peripheral |
| | | B: Within-group vigilance when surrounded/central |
| **Compatibility factors (feeding rate/food items and specific behaviours)** | | |
| 4 | Amount eaten + Food item + Behaviour + Age-sex class | Looking readily used when compatible with feeding |
| 5 | Biting + Digging + Handling + Pick + Searching substrate + Give groom + Self-grooming + Receive groom + Chewing + Self scratch + Movement + Posture + Age-sex class | Looking readily used when cost-free or compatible with underlying behaviours |
| **Reactionary stimuli and risks** | | |
| 6 | Time since within-group aggression + Age-sex class + Behaviour + Visibility + Rank | Reactionary vigilance for within-group threats |
| 7 | Time since mating + Age-sex class + Behaviour | Social monitoring for mates and competitors |
| 8 | Time since adult female calls + Age-sex class + Behaviour | Multifunctional calls – social monitoring or reactionary vigilance for within or extra-group threats |
| 9 | Time since adult or adolescent male calls + Age-sex class + Behaviour + Visibility + Rank | Multifunctional calls – social monitoring or reactionary vigilance for within or extra-group threats |
| 10 | Time since active heterospecific encounter + Age-sex class + Behaviour + Visibility + Rank | Reactionary vigilance for predator/extra-group threats |
| 11 | Time since passive heterospecific encounter + Age-sex class + Behaviour | Reactionary vigilance for predator/extra-group threats |
| 12 | Time since dog encounter + Age-sex class + Behaviour + Visibility + Spatial position + Number of neighbours | Reactionary vigilance for predator/extra-group threats |
| 13 | Time since alarm + Age-sex class + Behaviour + Visibility + Spatial position + Number of neighbours | Reactionary vigilance for predator/extra-group threats |
| 14 | Time since encounter with another group + Age-sex class + Behaviour + Visibility + Spatial position + Rank + Number of neighbours | Reactionary vigilance for extra-group threats |
| **Within-group threats** | | |
| 15 | Number of within-group threats + Age-sex class + Behaviour + Visibility + Rank + Number of neighbours | Within-group (social) vigilance |
| **Pre-emptive risks (spatial position/cohesion and landscape of fear for external group threats)** | | |
| 16 | Spatial risk of encountering a leopard * (Number of neighbours + Spatial position + Behaviour) + Visibility + Rank + Age-sex class | Pre-emptive vigilance for predators |
| 17 | Categorical habitat type * (Number of neighbours + Spatial position + Behaviour) + Visibility + Rank + Age-sex class | Pre-emptive vigilance for predators |
| 18 | Continuous landscape familiarity * (Number of neighbours + Spatial position + Behaviour) + Visibility + Rank + Age-sex class | Pre-emptive vigilance in unfamiliar areas |
| 19 | Categorical landscape familiarity * (Number of neighbours + Spatial position + Behaviour) + Visibility + Rank + Age-sex class | Pre-emptive vigilance in unfamiliar areas |
| 20 | Spatial risk of encountering another baboon group * (Number of neighbours + Spatial position + Behaviour) + Visibility + Rank + Age-sex class | Pre-emptive vigilance for other baboon groups |
| **Observer risks** | | |
| 21 | Observer tolerance * (Observer distance + Observer movement + Behaviour) + Age-sex class | Pre-emptive vigilance for observer threats |

overlapping zero (Supplementary Table S10). Looking duration was positively associated with chewing, but negatively associated with biting, digging, searching substrate, giving grooming, self-grooming, receiving grooming, handling, picking, and movement (Supplementary Table 10). These results highlight the constraints certain behaviours (e.g., grooming, digging) have on looking patterns, but also highlight the compatibility that biting, picking, and handling food items has with frequent but brief-looking bouts. Time spent resting was negatively associated with looking frequency and positively associated with looking duration (resting-only model: Supplementary Table S11; full model with multi-collinearity issues: Supplementary Table S12), highlighting that these animals maximised the duration of looking when it was cost-free.

**Reactionary risk models (models 6–14)**

None of the models incorporating factors on immediate risks to animals (reactionary risk models) produced substantial $R^2$ values or held considerable weight for looking frequency. In contrast, for the duration response variable, time since male vocalisations and extra-group/within-species encounters had stacking weights greater than 0.1 in both stacks, whilst all reactionary models had marginal $R^2$ values greater than 0.535. The total duration of looking was greatest whilst within-group aggressions, alarms, wahoos, and extra-group/within-species encounters were ongoing, but was relatively consistent across the remaining time categories, including when no event had occurred (Fig. 3). This suggests the study animals had a strong reactionary vigilance response to these stimuli but reverted to typical

**Table 2 | Stacking weights and Bayesian $R^2$ (credible interval in parentheses) estimates for models exploring the hypothesized drivers of frequency and total duration of looking bouts**

| # | Model | Frequency of looking bouts | | | | Total duration of looking bouts | | | |
|---|-------|---------|----------------|-------------|----------------|---------|----------------|-------------|----------------|
| | | Weights | Shared weights | Marginal $R^2$ | Conditional $R^2$ | Weights | Shared weights | Marginal $R^2$ | Conditional $R^2$ |
| 1 | ~1 | 0.076 | 0.031 | −0.011 (−0.014, −0.009) | 0.036 (0.019, 0.052) | 0.002 | 0 | −0.006 (−0.010, −0.003) | 0.039 (0.018, 0.059) |
| 2 | Age-sex class + Behaviour (Minimal) | 0 | | 0.018 (0.007, 0.031) | 0.052 (0.032, 0.070) | 0.044 | 0 | 0.537 (0.503, 0.568) | 0.539 (0.507, 0.572) |
| | **Group geometry and cohesion** | | | | | | | | |
| 3 | Number of neighbours * Spatial position + Age-sex class + Behaviour | 0 | | 0.032 (0.019, 0.046) | 0.061 (0.040, 0.080) | 0 | | 0.536 (0.503, 0.568) | 0.538 (0.506, 0.572) |
| | **Compatibility factors (feeding rate/food items and specific behaviours)** | | | | | | | | |
| 4 | Amount eaten + Food item + Behaviour + Age-sex class | **0.143** | **0.315** | 0.361 (0.324, 0.392) | 0.378 (0.345, 0.412) | **0.13** | **0.206** | 0.616 (0.581, 0.647) | 0.619 (0.586, 0.650) |
| 5 | Biting + Digging + Handling + Pick + Searching substrate + Give groom + Self-grooming + Receive groom + Chewing + Self scratch + Movement + Posture + Age-sex class | **0.242** | **0.548** | 0.354 (0.320, 0.385) | 0.382 (0.347, 0.413) | **0.101** | **0.259** | 0.682 (0.653, 0.711) | 0.686 (0.659, 0.715) |
| | **Reactionary stimuli and risks** | | | | | | | | |
| 6 | Time since Aggression + Age-sex class + Behaviour + Visibility + Rank | 0.023 | 0.07 | 0.023 (0.009, 0.036) | 0.054 (0.034, 0.073) | 0.08 | 0 | 0.549 (0.517, 0.580) | 0.551 (0.517, 0.582) |
| 7 | Time since Mating + Age-sex class + Behaviour | 0 | | 0.017 (0.004, 0.028) | 0.050 (0.030, 0.069) | 0 | | 0.535 (0.502, 0.568) | 0.539 (0.504, 0.570) |
| 8 | Time since adult female calls + Age-sex class + Behaviour | 0.038 | 0.004 | 0.020 (0.008, 0.033) | 0.055 (0.037, 0.075) | 0 | | 0.535 (0.502, 0.568) | 0.538 (0.506, 0.571) |
| 9 | Time since adult or adolescent male calls + Age-sex class + Behaviour + Visibility + Rank | 0 | | 0.023 (0.011, 0.038) | 0.052 (0.034, 0.072) | **0.114** | **0.251** | 0.552 (0.517, 0.583) | 0.554 (0.524, 0.587) |
| 10 | Time since active heterospecific encounter + Age-sex class + Behaviour + Visibility + Rank | 0.042 | 0 | 0.018 (0.004, 0.031) | 0.052 (0.033, 0.071) | 0 | | 0.537 (0.505, 0.572) | 0.539 (0.507, 0.573) |
| 11 | Time since passive heterospecific encounter + Age-sex class + Behaviour | 0.04 | 0.001 | 0.019 (0.006, 0.032) | 0.052 (0.033, 0.071) | 0.011 | 0 | 0.549 (0.518, 0.580) | 0.553 (0.520, 0.584) |
| 12 | Time since dog encounter + Age-sex class + Behaviour + Visibility + Spatial position + Number of neighbours | 0 | | 0.030 (0.016, 0.044) | 0.061 (0.039, 0.080) | 0 | | 0.548 (0.514, 0.578) | 0.551 (0.519, 0.581) |
| 13 | Time since alarm + Age-sex class + Behaviour + Visibility + Spatial position + Number of neighbours | 0 | | 0.031 (0.017, 0.045) | 0.059 (0.038, 0.078) | 0 | | 0.547 (0.515, 0.581) | 0.550 (0.517, 0.583) |
| 14 | Time since WE + Age-sex class + Behaviour + Visibility + Spatial position + Rank + Number of neighbours | 0.024 | 0 | 0.034 (0.018, 0.049) | 0.064 (0.044, 0.085) | **0.178** | **0.183** | 0.553 (0.521, 0.584) | 0.555 (0.522, 0.584) |
| | **Within-group threats** | | | | | | | | |
| 15 | WGT + Age-sex class + Behaviour + Visibility + Rank + Number of neighbours | **0.114** | 0 | 0.037 (0.022, 0.051) | 0.066 (0.045, 0.086) | **0.186** | 0.072 | 0.550 (0.517, 0.581) | 0.553 (0.519, 0.584) |
| | **Pre-emptive risks (spatial position/cohesion and landscape of fear for external group threats)** | | | | | | | | |
| 16 | Leopard RSF *(Number of neighbours + Spatial position + Behaviour) + Visibility + Rank + Age-sex class | 0 | | 0.032 (0.017, 0.047) | 0.062 (0.042, 0.081) | 0 | | 0.549 (0.517, 0.582) | 0.552 (0.519, 0.584) |
| 17 | Habitat type *(Number of neighbours + Spatial position + Behaviour) + Visibility + Rank + Age-sex class | **0.117** | 0 | 0.055 (0.037, 0.073) | 0.084 (0.060, 0.105) | 0.055 | 0.02 | 0.549 (0.516, 0.581) | 0.552 (0.519, 0.583) |

**Table 2 (continued) | Stacking weights and Bayesian R² (credible interval in parentheses) estimates for models exploring the hypothesized drivers of frequency and total duration of looking bouts**

| Model | Frequency of looking bouts | | | | Total duration of looking bouts | | | |
|---|---|---|---|---|---|---|---|---|
| | Weights | Shared weights | Marginal $R^2$ | Conditional $R^2$ | Weights | Shared weights | Marginal $R^2$ | Conditional $R^2$ |
| 18 Inverted UD *(Number of neighbours + Spatial position + Behaviour) + Visibility + Rank + Age-sex class | 0 | | 0.029 (0.014, 0.044) | 0.062 (0.042, 0.083) | 0.003 | 0 | 0.549 (0.516, 0.580) | 0.552 (0.520, 0.583) |
| 19 CFB *(Number of neighbours + Spatial position + Behaviour) + Visibility + Rank + Age-sex class | **0.175** | 0.031 | 0.032 (0.017, 0.048) | 0.065 (0.044, 0.086) | 0.088 | 0.027 | 0.548 (0.513, 0.579) | 0.550 (0.520, 0.582) |
| 20 Spatial WE *(Number of neighbours + Spatial position + Behaviour) + Visibility + Rank + Age-sex class | 0 | | 0.033 (0.020, 0.049) | 0.063 (0.043, 0.084) | 0.01 | 0.001 | 0.550 (0.518, 0.582) | 0.552 (0.520, 0.583) |
| **Observer risks** | | | | | | | | |
| 21 Observer tolerance * (Observer distance + Observer movement + Behaviour) + Age-sex class | 0 | | 0.021 (0.008, 0.034) | 0.055 (0.036, 0.074) | 0 | 0 | 0.535 (0.502, 0.568) | 0.538 (0.502, 0.569) |

Weights closer to zero indicate lower predictive accuracy of a model. A weight equal to 1 would indicate that a model predicts every data point with more accuracy that the other models within the stack. Weights in bold highlight values above 0.1. For frequency models, behaviour is the total time spent in engaged behaviours. For duration models, behaviour is the total time spent not in engaged behaviours. WE = within-species/extra-group encounter, either time since an encounter or spatial risk of encountering another group/foreign individual. WGT = count of within-group threats within 5 m of the focal animal, RSF = leopard resource selection function value at the location of the focal observation. UD = the utilisation distribution (of the baboon's home range) value at the location of the focal observation. CFB = if the focal observation occurred in the core, frequently used, or boundary of the home range. Marginal $R^2$ considers only the variance of the population-level (i.e., fixed) effects, whilst conditional $R^2$ takes both the population- and group-level (i.e., random) effects into account.

behavioural patterns very quickly; however, given the stacking weights, wahoos and encounters with extra-group baboons predict the duration of looking with more accuracy than within-group aggressions or predator-associated alarms, but less than the feeding rate/food items and specific behaviours models. See Supplementary Tables S13 and S14 for summary results for all reactionary models.

**Within-group risk (model 15)**
For both frequency and duration models, we found positive relationships between proximity to social threats and looking (see Fig. 4c, d, and Supplementary Tables S15 and S16). Interestingly, this relationship is opposite to what we found with looking patterns and proximity to group members generally (e.g., the 'many eyes' hypothesis) in the same models (see Fig. 4a, b). As the social threats variable was tuned to each individual, these results indicate that the focal animals were attentive to the identity of their neighbours and increased looking if their individual-specific risk increased, regardless of the potential risk reduction experienced with more neighbours (i.e., dilution or confusion effects). It should be noted, however, that the $R^2$ value of the frequency model is close to zero, suggesting it was poor at predicting new observations, whereas the duration model performed better.

**Pre-emptive risk (models 16–20) and observer effects (model 21)**
All pre-emptive risk models investigating the frequency of looking produced low $R^2$ values, with leopard risk (model 16 – leopard resource selection function value at the location of the focal observation), home-range familiarity (model 18 – the utilisation distribution value at the location of the focal observation), and spatial risk of encountering another group (model 20 – derived from the spatial distribution of interactions with other groups) models all holding zero weight in the initial stack. Frequency models for habitat type (model 17) and home-range familiarity (model 19) held weights greater than 0.1 in the initial stack but 0 and 0.031 respectively in the second stack. Yet, duration models produced $R^2$ values greater than 0.5, indicating moderate predictive performance; however, the leopard model held zero weight and the other models held minimal weight in the initial stack, with all zero or close to zero in the second stack. We did observe some minor relationships across the frequency and duration models which both support and oppose certain vigilance hypotheses (see Supplementary Tables S17–S26 and Figs. S1–S4). For example, being peripheral whilst at farms (where there is risk of being killed by humans) was associated with looking more frequently (see Supplementary Fig. S1). Models incorporating data on individual visual tolerance of observers and observer behaviours (model 21, see Supplementary Tables S27–S28) also exhibited poor predictive accuracy. This suggests observers were not a consistent nor significant driver of the focal animal's looking patterns.

## Discussion
Vigilance has long been considered one of the most ubiquitous anti-predator behaviours, yet there is generally a lack of evidence supporting the notion that animals must be vigilant to detect predators[23]. We used the looking framework, whereby all looking behaviours are sampled regardless of the underlying behaviours being performed, to identify the most prominent factors regulating the visual behaviours of a wild group of habituated chacma baboons. The looking framework allowed the data to reveal reactionary vigilance for external threats (e.g., other groups) and within-group (social) vigilance use in reactionary and pre-emptive scenarios, whilst still highlighting that specific behaviours and foraging tasks were the central drivers promoting and constraining looking.

We found that as the time spent biting, picking, and handling increased, the frequency of looking increased but the duration of looking decreased. We also found that items typically processed using their teeth (e.g., biting large fruits, seed pods, succulent leaves, and roots) were also associated with greater frequency of looking, whereas smaller items (e.g., small fruits and seeds, and invertebrates) were associated with less frequents bouts. When leaves, grass blades, and grass seeds were the predominant food item the duration of looking was greater than all other items, including when

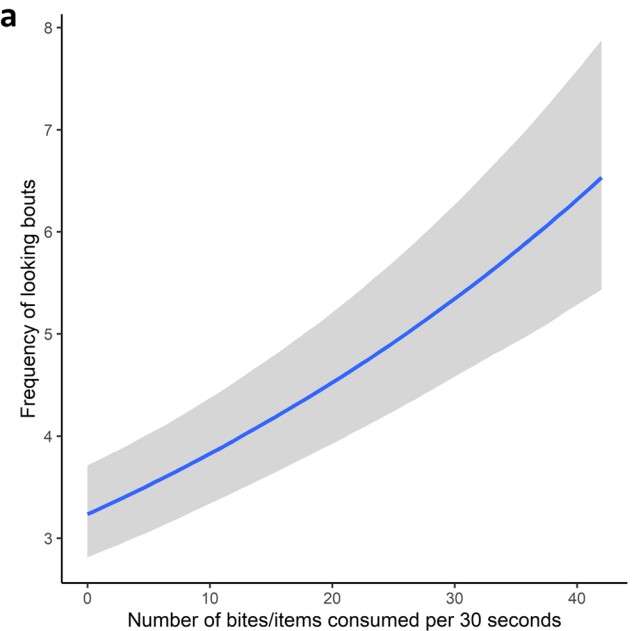
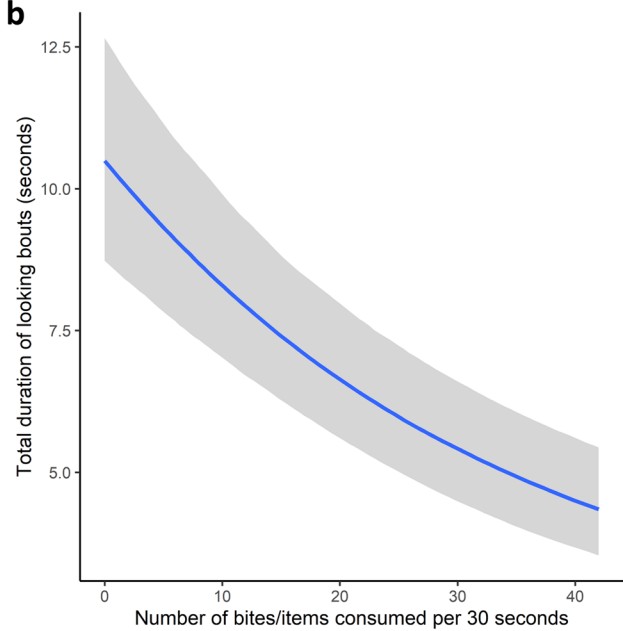

**Fig. 1 | Conditional effects plots displaying the relationship between looking patterns and the number of bites/items consumed within a focal observation.** **a** Relationship between the number of bites/items consumed within a focal observation and the frequency of looking bouts. **b** Relationship between the number of bites/items consumed within a focal observation and the duration of looking. Shaded areas display the relevant credible intervals (2.5% and 97.5% quantiles).

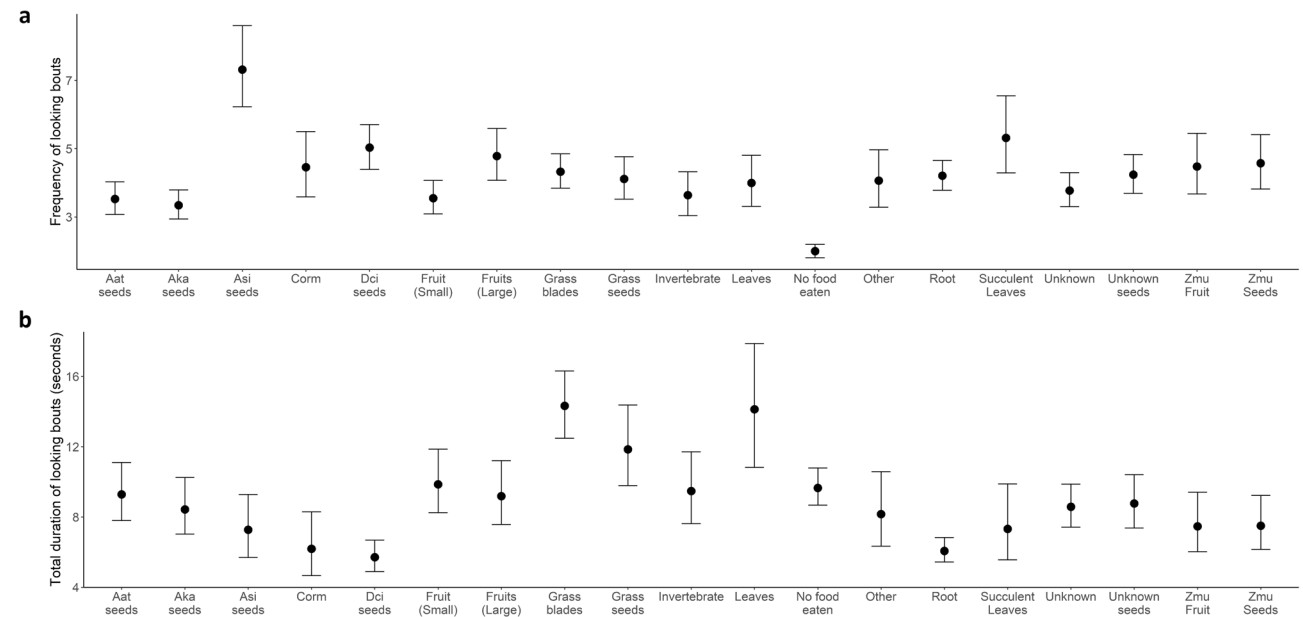

**Fig. 2 | Conditional effects plots displaying the relationship between looking patterns and foraging items.** **a** Relationship between the frequency of looking and the predominant foraging item searched for, manipulated/handled, or consumed (during a 30-s focal observation). **b** Relationship between the duration of looking and the predominant foraging item searched for, manipulated/handled, or consumed (during a 30-s focal observation). Dots display parameter estimates and bars display the relevant credible intervals (2.5% and 97.5% quantiles). Aat refers to *Acacia/Senegalia ataxacantha*, Aka: *Acacia/Vachellia karoo*, Asi: *Acacia/Vachellia sieberiana subsp. woodie*, Dci: *Dichrostachys cinerea subsp. Africana*, and Z.mu: *Ziziphus mucronata subsp. mucronata*, each of the species was commonly consumed and represented unique manipulation/handling tasks that were hypothesized to have differential impacts on looking patterns. Small fruits could be placed in a baboon's mouth whole whereas large fruits required several bites or manipulation. No food eaten refers to no food being consumed or foraged for during the focal observation. 'Other' were rarer items grouped together, including fungi, bamboo shoots, and animal matter. Succulent leaves included numerous Aloe sp. and *Opunita ficus-indica*. 'Unknown' was when the focal animal picked or consumed something the observer could not identify. Unknown seeds were seeds taken from the ground/leaf litter where it was clear seeds were being foraged but the precise identity of the species not known.

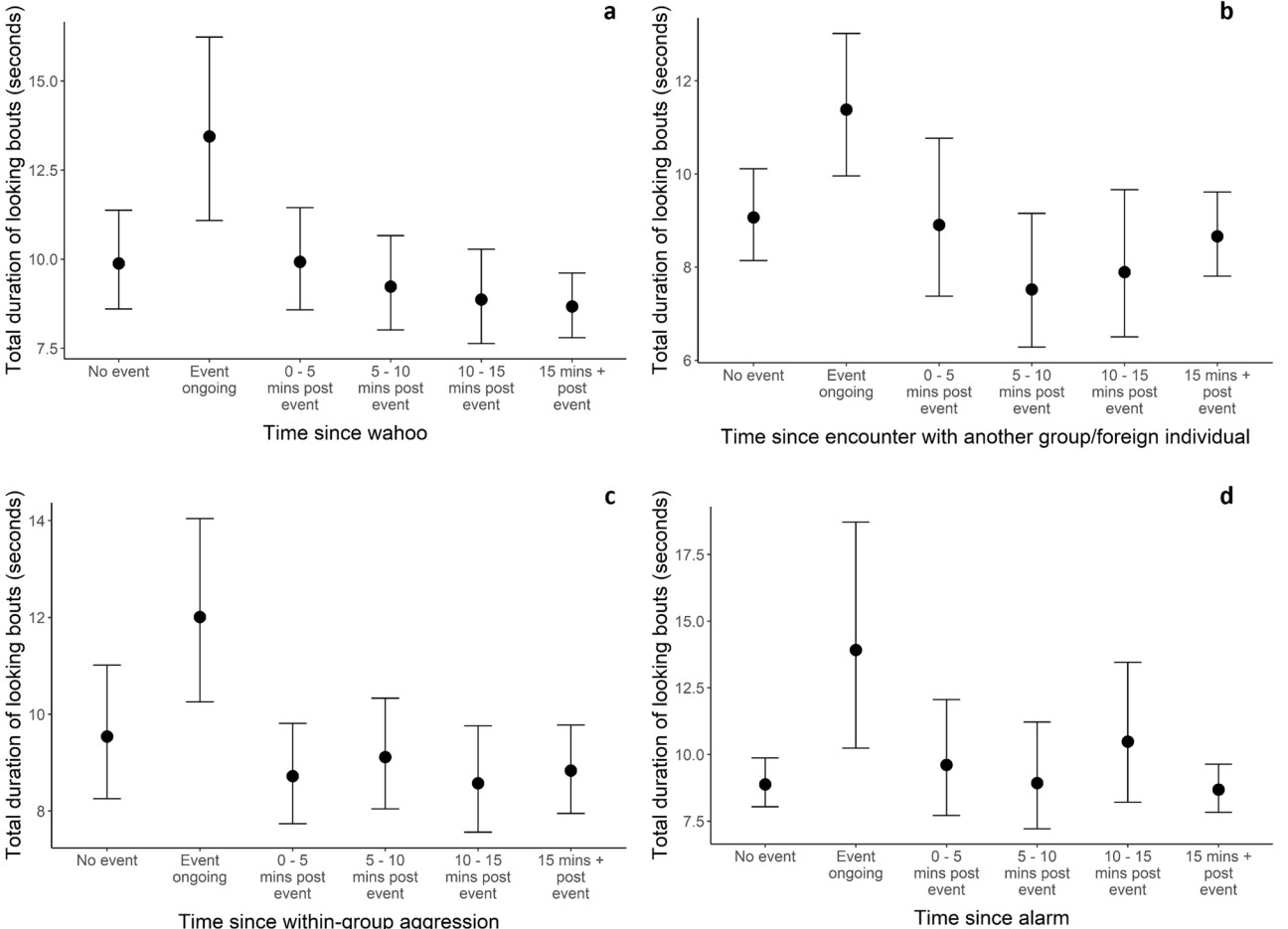

**Fig. 3 | Conditional effects plots displaying the relationship between the time since an event and the total duration of looking bouts. a** Relationship between the time since a wahoo and the duration of looking. **b** Relationship between the time since an encounter with another group/foreign individual and the duration of looking. **c** Relationship between the time since within-group aggression and the duration of looking. **d** Relationship between the time since an alarm and the duration of looking. Dots display parameter estimates and bars display the relevant credible intervals (2.5% and 97.5% quantiles).

no foraging or feeding behaviours took place, whilst food items with thick casings (e.g., seed pods) were associated with the lowest durations of looking (see Fig. 2). Digging and searching substrate time did not appear to influence the frequency of looking substantially but had a strong negative relationship with the duration of looking. These results align well with previous findings that vigilance use shares some compatibility with food handling in mammals[24–26] and birds[27], whilst tasks taking place exclusively on the ground (e.g., digging and searching leaf litter) can force animals into a head down posture, creating a trade-off between looking and foraging[28].

When taking the specific behavioural and food item results in combination with feeding rate results (i.e., feeding rate was positively associated with frequency and negatively associated with the duration of looking), our results support that tasks allowing for numerous moments of compatible-looking time may do so without sacrificing feeding rate significantly[18,26] (see Fig. 5). In fact, some feeding tasks may actively promote looking. For example, leaves, grasses, and grass seeds were abundant in most food patches such that picking behaviours did not seem to require a precise focus of attention and an animal could look towards the next food item, promoting looking, whilst continuing to pick. Thus, baboons seem to prioritise feeding over longer-looking episodes but have a consistent tendency to use the compatible and cost-free moments of their underlying behaviours to update their information on their surrounding environment[18], regardless of the current pre-emptive risk scenario. Investigating specific behaviours and foraging tasks in combination with feeding rate should therefore be especially important for research on species that use their hands to forage for or manipulate food. In addition,

given that birds can also raise their heads whilst handling food, resulting in a positive association between peck rate and predator detection[27], our methodological and analytical approach may reveal new insights into the contribution of vigilance to the general behavioural patterns of numerous taxa, potentially altering our concepts about vigilance and fear in the animal world.

Despite not attempting to sample vigilance specifically, we identified reactionary vigilance use (increased duration of looking) during periods of increased within-group conflict, wahoos, alarms, and encounters with other groups or foreign lone individuals. Interestingly, in all cases the animals returned to baseline levels of looking within 5 min, suggesting vigilance may often be a more induced behaviour in this group[18], and that if a threat is worth monitoring, the animals typically focus on it entirely, as opposed to utilising more frequent bouts or glancing.

Count of social threats within 5 meters was the only risk variable to produce the same positive relationship across both response variables. This supports within-group (social) vigilance hypotheses, which are also well supported across primate vigilance research[2]. Interestingly, the count of conspecifics within 5 meters had a negative effect on both looking variables, suggesting these animals perceived less risk from external threats (e.g., predators) when spatial cohesion was high[29], but altered strategies readily if their personal social risk increased. However, social threat models did not yield any weight in the simplified stacks and the $R^2$ value of the frequency model was low. It is therefore likely that the within-group threats model is good at predicting the duration of looking bouts when the number of threats is high but does poorly when they are absent. It seems likely that the baboons

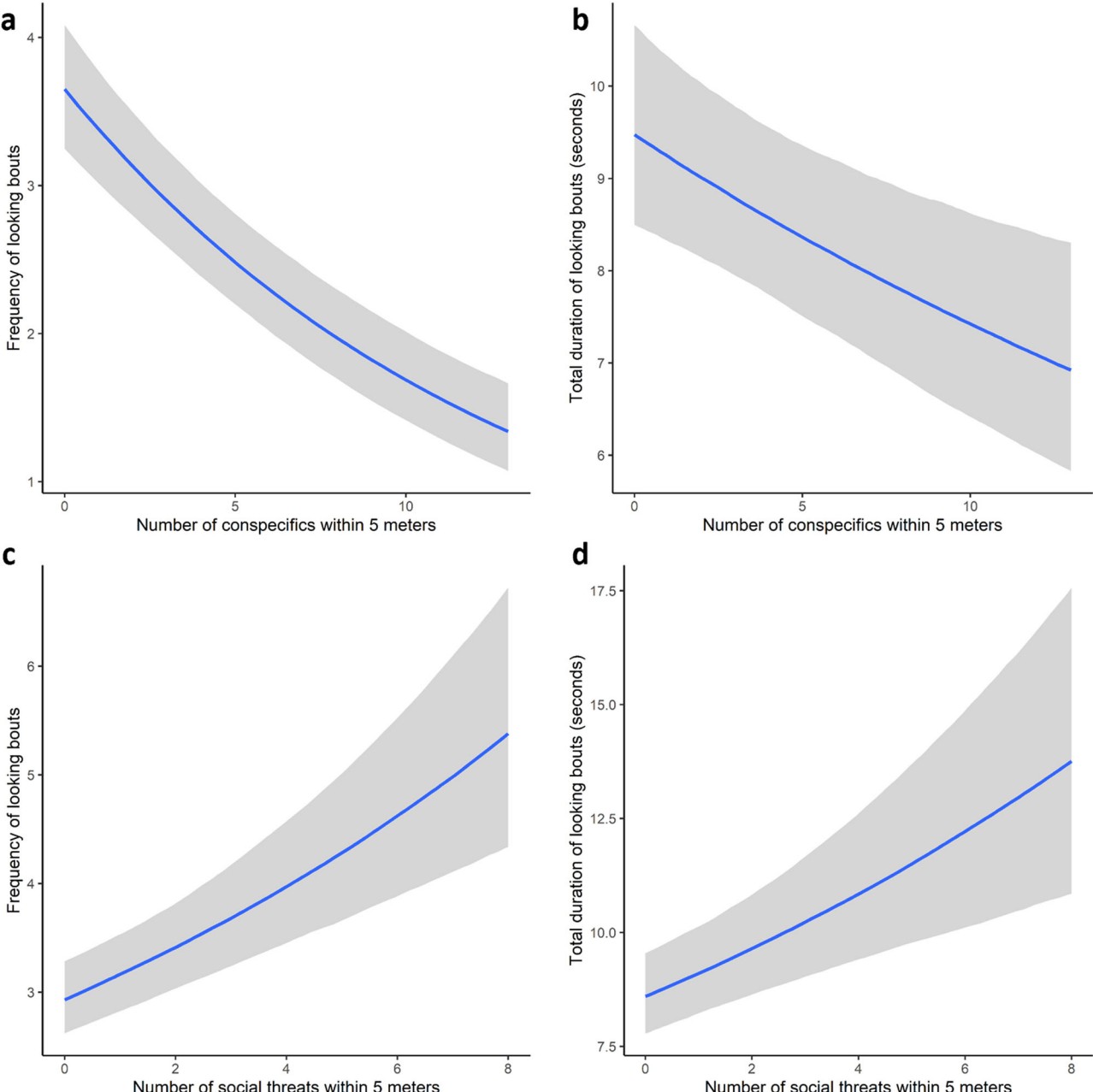

**Fig. 4 | Conditional effects plots displaying the relationship between looking patterns and the number of conspecifics and number of social threats.** **a** Relationship between the number of conspecifics (neighbours) within 5 meters and the frequency of looking. **b** Relationship between the number of conspecifics (neighbours) within 5 meters and the total duration of looking. **c** Relationship between the number of social threats within 5 meters and the frequency of looking. **d** Relationship between the number of social threats within 5 meters and the total duration of looking. Shaded areas display the relevant credible intervals (2.5% and 97.5% quantiles).

actively avoided spending considerable time near within-group threats[30], thus minimising the likelihood of being attacked and the need for pre-emptive within-group (social) vigilance. As intra-clique aggressions were low in our study group, we did not explore whether the number of high-ranking neighbours (including clique members) had an effect on looking patterns, but this may be appropriate in other groups and systems. It would also be interesting to explore how looking patterns are affected by the presence/number of social threats interacted with the number of neighbouring clique members as this would identify whether affiliates can buffer social threat perception.

The models specifically exploring the interaction between spatial position and cohesion garnered no weight in any of the stacks, again suggesting that broad risk dilution and confusion hypotheses concerning external threats (i.e. [8,9,31]) were not key drivers of looking. Future research may want to consider exploring the effect of 'isolation' explicitly however (e.g., no conspecific within 50 meters), as in our approach 'peripheral' observations still allowed for neighbours to be present nearby, which may have decreased risk perception. We also found little evidence that looking patterns were altered according to landscape familiarity or the spatial risk of encountering leopards or other groups, counter to findings supporting landscape of fear findings elsewhere[32] and in this group[33]. The contrasting results to the latter study is intriguing; however, despite the lead researchers and study animals remaining consistent across studies, a 'scanning' definition[34] and instantaneous point sampling protocol were used in the other study, reinforcing the notion that methodological consistencies are clearly needed when making comparisons[2,35,36], even within the same

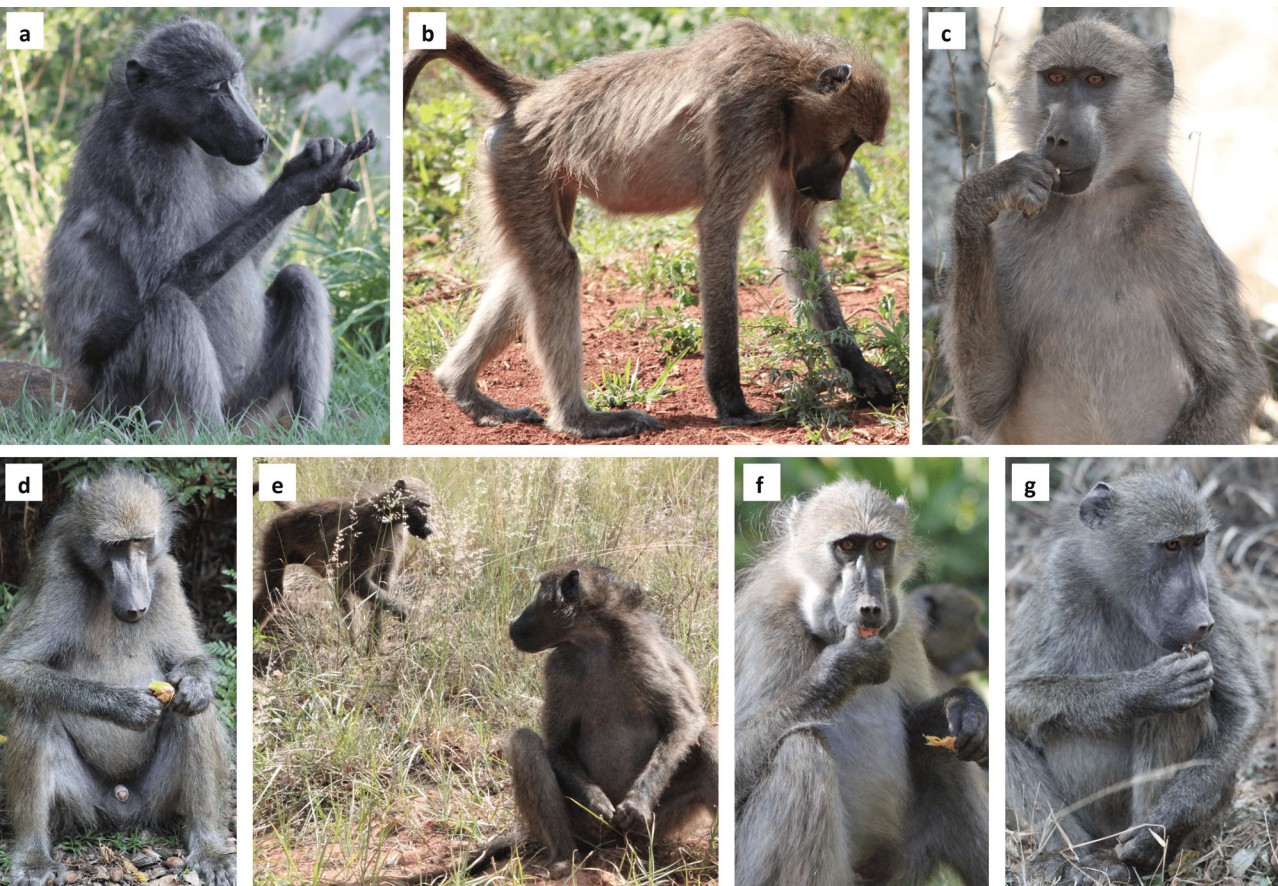

**Fig. 5 | Baboon behaviours and feeding tasks vary in their compatibilities with looking. a** Self-grooming and allogrooming were associated with lower durations of looking. **b** Digging and searching the substrate for food items were associated with lower durations of looking. **c** Picking was associated with more frequent bouts of looking. **d** Although handling food items was associated with lower durations of looking, these tasks could promote momentary opportunities for looking that could be used to monitor within-group threats, as shown in (**e**). Items processed using the teeth, such as large fruits (as shown in **f**), corms, succulent leaves, and seeds (especially from *Acacia sieberiana* subsp. *woodie* and *Dichrostachys cinerea* subsp. *Africana* – as shown in **g**) were associated with a greater frequency of looking bouts.

study group[19]. Lack of support for pre-emptive external risk hypotheses may be linked to the large group-size, whereby dilution and confusion effects are maximised, leading to reduced individual risk perception (for external threats). It could also be argued that there may be a human-shield effect present[37], i.e., that observers are consistently displacing predators such as leopards, artificially reducing the baboons' perceived predation risk[38], leading to diminished natural anti-predator behaviours. We do not rule out this possibility and encourage future research to explore human-shield factors explicitly, likely via comparative analyses making use of remote technologies[39].

Future research may also consider expanding our framework to include random slopes (e.g., spatial position over individual identity). This will reveal whether individuals differ in their looking responses according to various ecological scenarios (e.g., certain individuals may be more vigilant when in less accustomed spatial positions). Additionally, researchers may wish to incorporate additional interaction terms (e.g., age-sex class * spatial cohesion * spatial position or number of social threats * specific behaviours), as this may offer insights into nuanced risk hypotheses and their contribution to activity patterns generally. We don't dispute this, but increased sampling demands and analytical complexities involved may make such approaches challenging. The benefit of our approach is that the sampling requirements are feasible for most systems and allow for the most fundamental hypotheses to be investigated both independently and jointly in a consolidated approach. Thus, even if within-group (social) vigilance is more pronounced for specific foraging tasks, our current analysis would highlight this via stacking weights being shared between the within-group threats and specific behaviours models (i.e., both models would predict

looking patterns with similar precision). Such outcomes may then motivate additional model comparisons. Here, our results demonstrate that specific behaviours and feeding rate/food items independently predict the frequency and duration of looking with greater precision than risk models (that also include behaviour as additive effects); thus, more nuanced models are unnecessary.

It may be debatable whether animals should be attentive to the true spatial likelihood of encountering external threats or instead use spatial memory of encounters to inform their risk-sensitive behaviours. For example, the baboons may avoid areas where predator encounters have recently occurred, as opposed to increasing their pre-emptive risk-sensitive behaviours in areas where encounters are most likely (based on the previous movements of predators). The group's landscapes of fear could also be more sensitive to the interaction between predator behaviour and climatic conditions, e.g., misty conditions reduce visibility and are associated within increased leopard activity at Lajuma[33]. Future research should consider constructing landscapes of fear on multiple predators (and potentially integrating them into a single layer or variable) across various timescales and conditions as this should help elucidate important information about how animals perceive risk innately, the extent to which they can learn and adjust behaviours from experience, and the duration and extent of their spatial memory for threats.

The study group's looking patterns were also not consistently explained by the interaction between individual tolerance estimates and the proximity and behaviour of researchers. This finding adds some validity to our results but may also demonstrate that when aware of tolerance factors, researchers can adjust their behaviour accordingly (e.g., increase

observation distances for intolerant individuals), thus achieving the goal of having minimal influence on the behaviours we record. It also highlights that concentrating purely on focal animals as a measure for observer effects is inadequate, as across the same focal observations we showed that when the observer was within 4.5 meters (of a focal animal) that intolerant group members were less likely to occur in proximity and make physical contact with the focal animal[40]. Thus, despite minimising observer effects on the behaviours of focal animals, our presence may have altered the social environment for focal animals by buffering or displacing potential threatening conspecifics away from the focal animals. This is especially important to consider in other study systems where animals may be less habituated to observers.

To conclude, our analysis suggests that while elevated-looking patterns appear to be driven by vigilance use in certain reactionary circumstances, pre-emptive vigilance was not a consistent functional determinant of this group's looking patterns. Instead, the baboons seem to rely on the compatibility their natural behaviours have with looking and their capacity to collect multiple types of information concurrently to detect threats[14]. If this is the case, it could mean that any factor that encourages looking will increase the likelihood of the baboon's detecting a threat early[34]. Understanding the sensory capacity and threat detection capabilities of study animals should therefore be a topic for future research to explore in more detail as there is likely a differential need for pre-emptive vigilance across species and taxa with varying detective capabilities. Given the relationships we found for specific behaviours, there's a strong possibility that risk sensitivities may be very nuanced and therefore require these questions to be explored on finer scales. For example, future work could break down foraging behaviours into specific components (e.g., biting, pecking, digging, handling) and explore risk sensitivities within each specific behavioural bout. Such an approach would build a more complete picture of the compatible-looking time various species have according to the behaviours and tasks they engage in, and how such factors vary temporally, i.e., different seasons offer different foraging tasks.

It could be argued that research can adequately sample the various subcomponents of vigilance directly, e.g., routine/induced[41], pre-emptive/reactionary[42]; however, there is very little empirical evidence that researchers are able to do this task flawlessly. The looking definition and framework alleviate this issue and may improve inter-observer reliability[2,19]. A major criticism of such an approach may have been that it makes no attempt to sample vigilance specifically; however, the results of this study give support to the notion that risk-sensitive behaviours and their drivers can still be identified when using a broad definition and framework.

As little work has previously investigated so many competing risk drivers in combination with precise behavioural and task information, it may be that the contribution of 'vigilance' to looking patterns and overall activity budgets has been overestimated in many studies and systems. This would be especially true if animals prioritise alternative anti-predator/threat strategies (over vigilance) to minimize the risk of being detected and targeted (e.g., adjusting spatial position, avoiding risky places, staying near refuges, or readily using compatible-looking time during feeding or foraging tasks). Considering that the looking definition shares similarities to 'vigilance' definitions used in other studies and species, these findings are applicable to numerous taxa. We therefore encourage researchers to consider consolidating on similar definitions and adopting research frameworks that explore all major hypotheses in unison, this will allow researchers to tease apart the relative contributions of competing hypotheses to looking patterns, improving our understanding of the contribution of fear and vigilance to animal lives.

## Methods
### Ethical approval and permissions
All research methods included in this study were performed in accordance with the relevant guidelines and regulations, under ZA/LP/81996 research permit (Limpopo Province Department of Economic Development and Tourism), with ethical approval from the Animal Welfare Ethical Review Board at Durham University. The authors confirm they have complied with all relevant ethical regulations for animal use and the study was carried out in compliance with ARRIVE guidelines.

### Study area and group
Data were collected on a wild habituated group of chacma baboons (*Papio ursinus griseipes*) at Lajuma Research Centre, western Soutpansberg Mountains, South Africa (central coordinates S29.44031°, E23.02217°) between May 2018 and July 2019. The area was designated Afro-montane mist-belt community and contained a diverse range of natural habitats varying in plant species composition, canopy height, and foliage density[43,44]. Most of the study area was classified as a private nature reserve, but agricultural practices and habitat modification occurred in areas adjacent to the study group's core home range[45]. The major predator of baboons was leopards (*Panthera pardus*). The group were habituated for research purposes in 2005[14] and contained 80 individuals at the start of the study, increasing to 92 individuals by the end due to births (no permanent immigration took place during the study). From February 2015, ATLA typically followed this group between two and four days a week throughout the year and was proficient at identifying all individuals (including infants), even at a distance with binoculars. To ensure the baboons still experienced natural encounters with predators, we tried to minimize the study group's contact with humans during non-follow days (e.g., non-observers scared them from camps) and limited the number of observers to three during this study (one or two observers was the most common). In total, 65 baboons were used for this analysis, representing all non-infant individuals present at the start of this study (Supplementary Text S2).

### Video sampling methodology
30-s continuous focal sampling was used to record the temporal organisation of looking behaviours[1,2,46] using a high-definition video camera (Panasonic HC-W580 Camcorder). All of the video focal sampling was conducted by ATLA. The 30 s focal duration was selected after undertaking a pilot study designed to identify the ideal methodology (Supplementary Text S3 and Figs. S5–S9). Each observation day was split into four seasonally adjusted time periods that each accounted for 25% of the day length. A 'randomly' generated observation list was created and focal individuals were then selected pseudo-randomly from this list by sampling the first individual encountered from the top 15 identities on the list (approximately 20% of the original group-size). Individuals were never sampled more than twice a day or more than once in a single time period per day. More generally, we also tried to avoid sampling individuals within a time period if they already had one more sample in this time period overall than 75% of the group. For example, if 60% of the group had only one sample in time period 1 (T1), then an individual who already had two samples in T1 could not receive a third sample until <15 individuals (20%) remained that required a second T1 observation. This approach created issues whereby the top 15 individuals on the 'random' focal list sometimes could not be sampled for multiple days until sampling had evened out, but this avoided over-sampling individuals in certain season-specific time periods. As a result, the individual sampling effort was relatively even across months (range: 1–3) and between years (2018: 29–32, 2019: 25–28). We completed 3676 focal observations across the study subjects (range: 54–59 per individual) that were used in this analysis. Data came from 78 observation days, with individuals generally receiving 14 observations per seasonally adjusted time period, mostly in the range of 13–15, but some errors in the field meant that a few individuals received as few as 12 and as many as 16 observations in some time periods, with one individual receiving 18 in T2.

Focal observations were deemed successful if at least 25 s of footage had at least 50% of the animal's face and one eye completely in view. Observations were aborted or discarded if more than 50% of the focal animal's face was out of sight for more than 5 s. In situations where the focal animal was clearly using their hands (e.g., grooming, foraging, handling), ATLA also made sure we keep at least the focal animal's hands in view but generally

attempted to capture a view of the entire animals plus ~2.5 meters of the environment around them. Where possible, ATLA tried to start focal observations from at least four meters, but the baboons could often adjust position to narrow or increase this distance (see also ref. 41). Observers never adjusted their position if a baboon reduced their distance to the observer – displacing away from a baboon may be interpreted as a subordinate gesture and potentially lead to habituation issues and 'problem' animals developing that direct aggression towards observers. After failed observations, ATLA would then adjust position and try to restart the focal observation, a process that was repeated a maximum of three times before moving to another individual from the list. The individual receiving the aborted focal would then be reintegrated at the end of the list. Animals that disappeared during the study period were removed from the main focal-looking analysis, but their influence on focal animals (i.e., as a neighbour) was still explored for the periods they were still in the group.

## Operationally defining looking and extracting data from videos

Media Player Classic (MPC-HC: Guliverkli project) was used to slow down and extract precise looking bout lengths from videos (video skip length could be reduced to 4 hundredths of a second). A single observer (ATLA) extracted all looking and behavioural data from videos, thus removing the possibility of interpretation effects (as described in ref. 19). Extracting data from videos was initiated during fieldwork (i.e., on non-observation days) but some of the data extraction was completed after fieldwork had been completed.

A looking bout began when the focal animal's eyes were open, and its line of vision extended beyond (or diverted away from) its hands and the substrate, animal, or object its hands were in contact with[2,19]. The substrate usually referred to the ground but could also include rocks or branches the baboons were sitting or standing on or moving across. A looking bout ended when the focal animal diverted its line of vision towards an item in contact with their own hands, such as the focal animal's own body, foraging substrate, the ground (or another substrate they are sitting or standing on), or another monkey; or the animal closed its eyes. When animals were in contact with or facing large objects within arm's reach (e.g., tree trunks, rocks, buildings etc.,), these objects were considered an extension of the substrate and the animal had to divert its line of vision away from its hands and the object to be considered looking. In cases where an individual dipped its head to the substrate/object (e.g., ground when biting grassroots), to water (e.g., drinking), or to another baboon (e.g., to bite an ectoparasite), then the observer assessed whether its line of vision extended beyond the conspecific or beyond an arm's reach of the surface of the water, substrate, or object. In these cases, an arm's 'reach' was adjusted according to the size of the focal animal. This definition allows passive or non-goal-oriented looking to be recorded, which may be excluded by some observers when implementing definitions that use terminology such as 'gazing' or 'scanning/searching directed beyond arm's reach' (see also ref. 19 for discussion). After extracting looking behaviours from video footage only one observation occurred with less than 25 s with at least 50% of the focal animal's face in sight (24.639 s), we retained this observation as 'time in sight' was controlled for analytically (see Model structures).

The key concept underpinning this definition is that any looking behaviour allows for concurrent threat detection, thus, as long as an animal's line of vision extends beyond its immediate vicinity and is not obscured (e.g., by vegetation or objects) then it should detect a threat if it is there, regardless of its precise focus of attention[34]. In previous work, we experimentally validated that this was the case (using the looking definition) in this study group[14,47] and recommend that similar work is done in other systems before implementing the looking definition and framework. The operational nature of the definition therefore should allow researchers of other taxa to adapt it accordingly (see also ref. 19 for discussion). Thus for species without hands or forward-facing eyes (e.g., most birds or deer), the validation experiments should instead focus on the line of sight part only (regardless of postures) and identify operationalised distances (e.g., a wing or neck's length) at which visual obstructions hinder threat detection[19].

## Contextual variables

Contextual factors were collected at the beginning and end of focal observations and used as predictors within a range of candidate models (see Table 1). These included the number and identity of all neighbours within 5 meters of the focal animal (spatial cohesion), the estimated visibility (percentage) to 5 meters in all directions from the focal animal (see Supplementary Text S4 and Fig. S10 for detailed methods), and the distance between the focal animal and the observer. These were averaged across the start and end assessments so that each focal observation had a single value for each variable.

Additional variables were recorded at the end of each focal observation. We recorded habitat type as one of forest, woodland, bush, grassland, rock, camps, farms, and roads (Supplementary Text S5). Cliffs were incorporated into these categories according to the underlying substrate or vegetation structure (e.g., rock, grassland). We did not investigate focal animal height as it was challenging to complete observations on animals high above the ground due to visibility, practicality, and safety concerns; thus, focal samples are biased towards locations relatively near to the ground. We also noted whether the observer moved at any point during the focal observation (e.g., to keep the animal in view).

Spatial position was assessed as whether the focal animal was within the centre or on the periphery of the group for the majority of the focal observation. An individual was peripheral if on the edge of the group or had no more than 5 non-infant individuals (~5% of the group) between itself and the edge of the group, based on sightings and audible cues given by other group members. This was assessed from ATLA's position when it was unambiguous (e.g., high visibility locations allow a good view of the entire group) but if visibility was an issue, then ATLA and the other observer(s) quickly assessed the broader area. Although five individuals may seem a large buffer that could dilute the risk of predation/attacks, inter-individual spacings were typically irregular. Thus, although five local individuals may be 'further away' from the rest of the group, it was rare that they were all directly 'outside' of the focal, and so would have little influence on dilution effects.

Reproductive information for the focal animal, including consortship information and female cycle status (e.g., sexual swelling present, not cycling, lactating/infant carrying, pregnant etc) was also recorded, along with the age-sex class and location (i.e., distance or out of sight of the mother) of the offspring of lactating/nursing females.

The duration of each behaviour exhibited by the baboons was extracted from the focal videos by ATLA. Engaged behaviours were those requiring visual attention and use of the hands, including grooming another individual, self-grooming, digging, searching substrate, and picking. Picking was the action of picking or pulling a food item towards their mouths and would often lead to the entire item being consumed without further processing. However, if the item was bitten or manipulated further, then the picking bout would end and a handling or biting bout would start. These behaviours were used as additive effects in the specific behaviours model (see Table 1: model 5) or grouped together as 'engaged' behaviours and included as a covariate (see Table 1: all models except 1 and 5). Aggression/play (fighting/wrestling and the aggressive/submissive vocalisations and screams during these episodes, chasing, pinning, biting, ground-swiping, threat/play facial gestures) were also recorded and considered engaged behaviours but were not investigated within the specific behaviour model as they were undersampled.

Not-engaged behaviours included resting, chewing, mating, self scratch, receiving grooming, drinking, movement (e.g., walking/running when not foraging or engaged in socialising and aggression/play), communication (e.g., facial gestures and greetings), biting, and handling. Biting was defined as when animals take several smaller bites of large food items, instead of placing whole items in their mouth (e.g., 'picking' small fig fruits). Handling involved the action of cleaning dirt off of roots or the use of their fingers to peel or pull open casings of some thick-skinned fruit or seed pods or pick off wings/legs of invertebrates. The food species and food items were recorded during the focal observations and the feeding rate (total number of

bites taken and items consumed during the observation) determined from video playback (see Table 1: model 4). Communicative gestures, drinking and mating observations were rare and so not included within the specific behaviour model but were accounted for in time spent not engaged calculations, which was included as a covariate in all models except 1 and 5 (see Table 1).

Finally, it was noted whether certain events were ongoing during each focal observation, including within-group events such as copulations and aggressions, and loud vocalisations, such as those made by females (e.g., lost calls) when the group was very dispersed and by males (i.e., wahoos) during a range of scenarios. Alarm calls were recorded as distinct to other vocalisations when cooccurring with certain behaviours (e.g., fleeing behaviours, screaming) or the threatening stimuli was identified (e.g., a leopard). Encounters with domestic dogs were coded separately to other factors. Encounters with other species were coded based on whether the event was considered passive or active. Passive encounters included other animals, such as bushbuck (*Tragelaphus scriptus*), coming within 10 meters of a group-member with no detectable behaviour change or interaction between the two species. Active encounters occurred when some form of displacement or agonistic interaction occurred between the two species (e.g., fighting was frequently observed with samango monkeys (*Cercopithecus albogularis schwarzi*)). We grouped passive (e.g., distant visual contact) and active (e.g., agonistic interactions) encounters with foreign baboons as all encounters elevated the threat level. Similarly, we did not distinguish between encounters with other baboon groups and encounters with foreign individuals, partly because it was sometimes hard to identify if an individual was truly alone, but also because encounters with lone males usually elicited group-wide alarms. The time each of these events occurred and appeared to end was recorded ad libitum throughout the day by ATLA and the other observers present, allowing us to calculate the time since each event as: no event (during the day so far), event ongoing, 0–5 min post event, 5–10 min post event, 10–15 min post-event, and greater than 15 min post-event.

### Calculating dominance rank and within-group threats

We recorded aggressions and displacement/supplant events ad libitum and created separate directed matrices for 2018 ($n = 638$ observations) and 2019 ($n = 695$). We then calculated the dominance rank for each year using the isi13 function from the 'compete' package[48]. Individual rank was then included as a covariate in several models (see Table 1). The dominance rank information was then applied to the identity of all neighbours within 5 meters of each focal observation, producing a count of higher-ranked neighbours. Since higher-ranking neighbours could be affiliated with the focal animal and unlikely to be considered threatening, we refined the number of social threats variable so that it was not biased by affiliates: i.e., the number of higher-ranked neighbours minus the number of higher-ranked clique members (see Table 1 – model 15). Clique membership was calculated using dyadic grooming data and community detection in igraph with the spinglass algorithm[49], see Supplementary Text S6 for details.

### Spatial variables for pre-emptive risk hypotheses

Between February 2015 and July 2019, ranging data were collected every 20 min during dawn-to-dusk follows of the study group ($n = 11,936$ GPS points) and encounter data for all interactions with other groups of baboons or foreign individuals ($n = 240$). We calculated a 99% utilisation distribution at 1% intervals via Time-Local Convex Hull Analysis[50], incorporating all ranging data from 2015 to 2019. To turn this utilisation distribution into a continuous home-range familiarity variable, we applied a linear stretch to rescale the utilisation distribution predicted values between 0 and 1[51]. We then inverted the scale so that the hypothesized positive relationship between risk and vigilance could be visualised appropriately. For the categorical variable (for home-range familiarity), we defined the isopleths at 33.3% intervals to explore whether distinct differences between core ($n = 1302$ focal observations), frequently used ($n = 1352$), and boundary areas ($n = 1022$) influenced looking patterns. We used the same methods (as with the utilisation distribution at 1% intervals) to calculate the distribution

of inter-group encounters during the same period. In this case, the time-scaled distance metric was set to 0 to reflect GPS points being collected opportunistically. The subsequent distribution was then scaled (as above) and divided by the scaled utilisation distribution to produce a layer providing a proxy for the spatial probability of encountering another group (offset by home-range utilisation), this variable was scaled a further time to ensure all values were between 0 and 1. To explore whether the study group altered their looking patterns pre-emptively in response to the spatial risk of encountering leopards, we used the scale integration[51] of the 2nd and 3rd order Resource Selection Functions (RSF) calculated by Ayers[33] for leopards occupying the same study area as the baboons - a proxy for the spatial probability of encountering a leopard (for further details see Supplementary Text S7 and Figs. S11–S15).

### Observer tolerance

We explored whether observer distance (to the focal animal) and movement during focal observations (coded yes/no) interacted with individual visual tolerance of observers to influence looking patterns (see Table 1 – model 21). Using Flight Initiation Distance methods, we previously quantified the distance at which each baboon visually oriented towards approaching observers, which was found to be distinct amongst individuals and consistent across time and scenarios[14,47]. To quantify each individual's visual tolerance of the observers, we extracted the individual-level effects (i.e., conditional modes) from a model exploring the visual orientation responses of the study animals to approaches made by observers (using the same data from ref. 14). See Supplementary Text S8 for complete methodology and Table S29 and Fig. S16 for the results associated with this analysis.

### Model structures

We examined the drivers of two dependent variables, the frequency and total duration of looking bouts within 30-s focal observations, in separate arrays of models, all fitted using the brm function from the 'brms' package[52]. As the focal animal's face could go out of sight temporarily, we included the duration of the observation that 50% of the animal's face was visible as an offset variable in all models. Duration models used a Gaussian family with an identity link and so the offset variable was not transformed. Since frequency models used a Poisson family with a log link, the natural log of the offset variable was used.

Observations were considered right censored when the total duration of looking was equal to the duration of time at least 50% of the animal's face was in view. This approach allowed the duration models to predict accurately beyond the 30-s cut-off imposed by the sampling design. As it is impossible for the duration to be less than 0, we defined a lower bound of 0 (i.e., truncated) to the posterior distribution to ensure data was modelled correctly. For all duration models, we allowed all parameters to be initialised at zero, allowing the no U-turn sampler to efficiently produce a finite log posterior[52].

Following the information-theoretic approach of Burnham et al.[21] we developed a series of models designed to weight the main theoretical drivers of looking (Table 1). Age-sex class and behaviour were included in all but the intercept-only model. In frequency models, the behaviour was the total time spent 'engaged' as these behaviours required the focal animal's focus of attention, and thus the frequency of looking was the more likely risk-sensitive behaviour. For duration models, the behaviour was the total time spent 'not engaged' as these behaviours did not require the animal's focus of attention, and thus the duration of looking was the more likely risk-sensitive behaviour. Reactionary variables (e.g., time since aggression) did not contain any interactions (e.g., with spatial position) as the ongoing event should be a clear driver to exhibit changes in looking duration or frequency regardless of the animal's current behaviour or scenario. Pre-emptive risk factors (e.g., spatial risk of encountering another group) should be more sensitive to behavioural and individual factors and therefore several 2-way interactions were included (e.g., the interactions between leopard RSF and spatial position, spatial cohesion, and current behaviour (time spent 'engaged' or 'not engaged'; see Table 1). Theoretically most hypotheses

could warrant 3-way interactions (or more) as well as random slopes over individual identity, but we did not pursue these options as the models would have become very complex and likely overparametrized/unreliable. We did not centre or scale any variables as several were categorical, had meaningful values for zero (e.g., time allocated to behaviour or number of neighbours), or were already scaled (leopard RSF, utilization distribution, and within-group encounter risk), whilst tolerance represented each individual's mean difference to the population mean. For all models, the observation date was also a random effect, crossed with individual identity.

For all models we used the default Student-t priors (df = 3, mean = 0, scaling factor = 10) for all model components. In the case of the standard deviations of group-level (i.e., random) effects, these parameters are constrained to be positive and therefore a half Student-t prior was implemented. All models produced high estimation accuracy, including at the tails of the distribution[53] (see Supplementary Text S9 for details on model checks). The variable 'time spent resting' created multi-collinearity issues in the specific behaviours models (model 5) and was therefore not included in the main analysis (Table 2), but results from a resting-only model are reported in the results.

### Assessing relative model prediction performance

We estimated the pointwise out-of-sample prediction accuracy from each model using leave-one-out cross-validation (LOO) from the 'loo' package[54]. LOO is computed via a Pareto smoothed importance sampling (PSIS) procedure for regularising importance weights[22]. PSIS approximation reliability was confirmed by inspecting the estimated shape parameter $\hat{k}$ diagnostic values in the generalized Pareto distribution, thus ensuring that extreme values are not too influential[22,55]. The LOO process uses n-1 sample points (focal observations) to tune a specific algorithm to predict the left-out point, allowing the n-1 samples to act as a training set for optimising the free parameters of the model and assess how well the tuned algorithm performs at predicting the left-out sample point. This process is repeated for the remaining samples and produces a test performance for all samples within each model. The resultant estimates therefore represent relative model prediction performance based on the full distribution of model parameters, unlike simpler estimates of predictive error (e.g., Akaike information criterion or deviance information criterion) that use only a single point estimate to approximate out-of-sample fit[56].

We employed Bayesian stacking using the loo_model_weights function from the 'loo' package. We developed a number of 'stacks' to explore the relative weighting of each hypothetical driver of looking. When comparing two (or more) models using stacking with PSIS-LOO values, stacking utilises the data produced from the PSIS-LOO procedures of each candidate model and compares the performance and accuracy of each model at predicting each left-out sampling point. Compared to other methods, such as weights produced from the widely appliable information criterion or pseudo-Bayesian Model Averaging, stacking performs well when several candidate models share similar covariates. This is achieved by optimizing the model weights jointly, allowing for similar models to share their weight whilst more unique models keep their original weights[22]. Thus, when a similar low stacking weight is shared across a number of models, it suggests that these models share similar prediction accuracy[55], i.e., certain drivers predict some sample points with accuracy but perform poorly at predicting looking behaviours across a broad range of scenarios. Therefore, a model with the maximum weight of 1 would predict every observation with the most accuracy, whilst a model with the lowest weight of 0 would not predict a single observation with better accuracy than any of the remaining candidate models. The initial stacks for each response variable contained all theoretical models in their respective stacks. A subsequent stack was then computed for each response variable including only models that shared non-zero weights in the initial stacks (Table 2). This was done to identify which model weights in the initial stack were derived from the joint optimization procedure combining weights of similar models[22]. We also calculated Bayesian $R^2$ estimates for each model using the r2_loo function from the 'performance' package[57], as a data-derived estimate of the proportion of variance explained for future observations predicted using a given model[58].

### Statistics and reproducibility

Each model used the complete dataset (n = 3676 total focal samples from 65 individuals, range: 54–59 focal samples per individual) and was run for four Hamiltonian Markov chains for 2000 iterations, with warmup iterations set to 1000, totalling 4000 post-warmup draws (default settings for brms). We used model estimates and 95% credible intervals to assess the effect of each predictor variable and interaction term on our response variables. If estimates were non-zero and credible intervals did not overlap zero, we inferred evidence that a predictor variable had an association with the response variable. Models were considered to be accurate at predicting looking patterns if their stacking weight was at least 0.1.

### Reporting summary

Further information on research design is available in the Nature Portfolio Reporting Summary linked to this article.

## Data availability
Raw data can be found on figshare (https://doi.org/10.6084/m9.figshare.26105674.v1)[59].

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

## Acknowledgements

We thank Prof. Ian Gaigher and Jabu Linden for permission to conduct research on the Lajuma property, and the neighbouring landowners for access to their properties for data collection. We thank two anonymous reviewers for their feedback which greatly improved the final version of the manuscript. We also thank the staff, assistants, and students at the Primate

& Predator Project and Lajuma Research Centre staff and students for some useful observations over the course of this study. ATLA was funded by a Natural Environment Research Council (NERC) studentship through the IAPETUS Doctoral Training Partnership (NE/L002590/1). This manuscript was additionally supported via a Leverhulme Early Career Research Fellowship awarded to ATLA (ECF-2023-318) in the Department of Anthropology, Durham University. We are grateful to Durham University, the Earthwatch Institute and an anonymous donor for funding the Primate & Predator Project's long-term data collection.

## Author contributions

A.T.L.A., L.R.L., and R.A.H. conceived the initial ideas. A.L.B., B.J., Z.M., T.P., F.S., A.W., A.F.W., and H.W. assisted with the evolution of the study and formulating the final goals, aims, and methods of the project. A.T.L.A., L.R.L., A.L.B., B.J., Z.M., T.P., F.S., A.W., A.F.W., and H.W. collected behavioural, ranging, and interaction data for the baboons. A.T.L.A. analysed the data. A.T.L.A. and L.R.L. drafted the initial manuscript. A.L.B., B.J., Z.M., T.P., F.S., A.W., A.F.W., H.W., and R.A.H. critically revised the manuscript. All authors gave final approval for publication and agree to be held accountable for the work performed therein.

## Competing interests

The authors declare no competing interests.
