## [Peer Review File · Communications Biology]

Reviewers' comments:

Reviewer #1 (Remarks to the Author):

The manuscript investigates vigilance behaviour within a (previously proposed) broader 'looking framework' that encompasses multiple competing hypotheses (social threat, tolerance to human observers and predator detection), using chacma baboons as a model system.

It is generally well-written, identifies an interesting, topical issue that merits publication and is an impressive analysis of a focused dataset. However, I did find that the description of data collection lacked clarity in many places. For example, several terms/categories are introduced in the main section and results that are not mentioned in the Methods, making tracking how data were collected and what is being analysed a little difficult. I had one issue with the analysis of social threat and the measure calculated. I hope the comments below will be helpful in improving the paper.

Introduction

Line 43 Identity = identify

Line 56 I think the term 'search' here is a little confusing (also line 20). As the authors' know, it wasn't specified in the Allan and Hill definition but was attributed to the Treves definition 'any visual search or scanning "directed beyond an arm's reach"'. The term search implies a goal oriented behaviour and subjectivity in deciding whether looking involves searching or not.

Results

Line 72 'foraging task' does not appear in the Methods or the supplementary materials making it difficult to match with data collected. 'Specific behaviours' also does not appear in the Methods/data collection protocols although it is mentioned in the model rationale in the supplementary. Please be consistent in use of terms throughout and define these categories in your data collection section. Similarly, within-group threats in referred to the next line, is this within-group risks in Table 1 (Model 15)?

Line 84-85 What is the difference between an extra-group member and a foreign individual, and if you make a distinction how did you distinguish between them?

Line 108 Table 1. The sub-heading 'Compatibility factors (specific behaviours and feeding rate)' is a little misleading as only Model 5 is referred to as the specific behaviours model. Perhaps a different sub-title would be appropriate here so that Model 4 with 'Behaviour' can be included. Is part of the description for Model 5 missing? 'Within-group risks' is separate from the sub-section of pre-emptive risks but included in pre-emptive vigilance in Table S2, can you re-name? Observer movement is missing from the Methods and only appears in tables. Can you please describe how it was collected? Please define 'vis' in the Table caption and use vis consistently throughout the table.

Line 126 Self-grooming, also referred to as auto grooming in your tables, please be consistent in terminology.

Discussion

Line 185 The initial summary a little vague and it would be useful to refer to the frequency and duration of vigilance here.

Lines 227-8 It's not clear here, or in this paragraph, if the authors are talking about frequency or duration of vigilance or both.

Line 242 Given that male baboons have often been found to often occupy the edges of the group it would have been interesting to see interaction effects with age-sex class for the pre-emptive models. Why wasn't this included and is this a limitation that you can discuss?

Line 246 Methodological consistencies may play a role but for data collected on the same group by some of the same researchers, this explanation is a little weak. The citation here is not published but under review – is this data on the group from the same time period? If so, what specifically about the analyses could account for these differences? Was the same definition of looking behaviour utilised? How were data collected differently. This is an important point and suggests that the findings here are

not necessarily robust, unless further explanation can be offered.

Line 278 From what you say above, this seems contradictory with your other research under review?

Line 298-9 The approach outlined, whilst all encompassing, reflects an enormous body of work from a large team of researchers. Recognition of the resources required to repeat such an approach is important here I think.

Methods

Line 310-313 Mention here that individuals were individually identifiable. This isn't explicitly stated but is then deduced further down where a focal list is created for observations.

Line 315 How did you arrive at 30-sec video samples?? Can you report looking distributions as the authors recommended in Allan and Hill 2017?

Line 316 What field of view (around the whole? animal or face) was recorded when focal observations were taken? This seems important considering the definition (see comments below about arm's reach). Was there a protocol for video-ing that included a minimum approach distance?

Line 317 'four time-periods' – did you attempt to sample equally within each time period and did you avoid sampling twice on the same day? The crossed random factors of date and ID suggest repeated sampling on the same day. If so, how many different observation days are represented in the 54-59 observations per individual? For example, are there 15 days of data for an individual with observations in each time of the four periods in one day? A better explanation of how the data were collected would help the reader understand the quantity (and reliability) of the dataset.

Line 327 Whilst the number of observations per individual ranged from 54-59, this constitutes 27 minutes per individual and 29-h of data for the group. Could you explain about how this was distributed over the 14-month fieldwork period? For example, was each individual sampled twice a month?

Line 332 I find the definition 'its line of vision extended beyond its hands and the substrate..' unclear, This could be taken to mean the substrate e.g. ground is always excluded from looking behaviour or only the immediate substrate, beyond the hands or in an area around the animal. Later on at line 337-40 it is partly clarified that large objects in arm's reach (trees, rocks) were considered substrate and so excluded. This means the line of vision must extend beyond arm's reach – can you add this to your definition? Does this also mean a tree or rock 1.5m away was included in looking? Line 335 also states that the animal 'had to divert its line of vision away from its hands'. In the definition, looking 'beyond,' could just mean looking past the hands and so the hands could be in line of sight – can you add that the line of vision had to be diverted from hands? The information at lines 337-40 needs to be incorporated into the definition as these criteria are crucial elements.

Line 346 It would help if 'spatial cohesion' could be added here as it took me a little time to search through to find out how cohesion was calculated.

Line 356 It seems the periphery of the group was determined through sightings and audible cues – were these collected by walking around the perceived group perimeter after each sample, or collected from the observer position after the video recording? If the latter, this would seem open to bias?

Additionally, as a sex bias has been noted in baboon individuals with males more likely at the front of the group e.g. van Schaik et al 2021, Fig.2 (DOI: 10.1111/eth.13233) and more likely engaged in vigilance (Matsumoto-Oda et al 2018) can you discuss what impact excluding this has on the results?

Line 360-79 Can you include and ethogram (descriptions) of your behaviours e.g. agonistic interactions span many types of behaviour in the supplementary? Similarly, movement can be collected in different ways.

Line 371 Not engaged implied no visual attention required, and in the model justification it was used as a predictor for duration of looking. Mating is included here but could refer to copulation (visual attention on the female?) or consortship (visual attention on the female and surroundings?). The rationale for including it here is not clear. Drinking behaviour is also classified as not engaged but would seem to involve head down and thus visual attention is occupied? Greater explanation in an ethogram might resolve this lack of clarity.

Line 380. After stating that events were recorded if ongoing, line 392 then says 'no event (during the day so far)' and that timing post-event was calculated– does that mean you also separately recorded all occurrences or ad lib events throughout the day? If so please add this sampling method.

Line 402 Number of social threats reflects the number of higher ranked non-clique neighbours but this metric doesn't distinguish between e.g. a situation where there is a single higher-ranking non-clique neighbour within 5m (a value of 1), and a situation where there are three higher-ranking neighbours including two higher-ranking clique neighbours (a value of 1) but the social threat level should be much higher in the first context. A better measure might be to divide your metric by the total number of higher ranked neighbours.

Line 430 Insert 'how'

Line 431 In the Supplementary S7, there is much reference to updating (47). Is this a reference – if so it's Vehtari, A., Gelman, A., Gabry, J. & Yao, Y. loo: Efficient leave-one-out cross-validation and WAIC for Bayesian models. R package version 2.0.0, which makes the methodology a little hard to follow. Can you also define FID at first mention before using the abbreviation.

Line 446 As Table S2 presents the hypotheses tested by the study I think it should be moved to the main part of the paper rather than in the supplementary (where detailed justification of the models is given).

Lines 459-63 This information is about creating your models and model structure so would be better placed up at line 446.

Supplementary

End of page 2, 'Each response variable also included a model (Model 5) designed..' there should be more model numbers referenced here?

Table S2 I think model 12 is missing the word 'neighbours' at the end? Model 7 is included in the sub-section Reactionary risks in Table 1 but there is no rationale for Model 7 in the text that follows or mention of reactionary vigilance.

Reviewer #2 (Remarks to the Author):

In this paper, the authors investigated the drivers of looking in a large troop of baboons. What is new here is the large number of variables included in the analysis including various families of drivers related to food, predation and social environment. The discussion of the findings is clear and illuminating. The results confirmed several patterns established in other species and uncovered novel ones like the effect of observers. The scope of the analysis is really broad and unique. I would like to raise some caveats about the analysis in the spirit of transparency. They should not be viewed as negative comments.

First, looking was investigated across the day and so presumably it includes episodes during which the animals were resting, socializing or foraging. The drivers of looking might not be the same during these various episodes. For instance, when feeding, looking might be affected by the sort of food eaten and less by social factors. Social factors might play a larger role when animals are resting or grooming. Much of the literature on vigilance typically focuses on specific contexts such as resting or foraging but rarely across contexts like in this study. Can this explain some of the findings?

Second, the analysis focuses on different families of drivers (geometric, within-group risks, reactionary risks, etc.). As far as I could tell, model comparisons did not consider cases mixing different families of risk factors. I understand that the paper already considered 21 models, but it should be worth mentioning that other models in the future could include combinations of various types of risk factors.

Third, again as far as I can see, model comparisons rested on predictive power alone (using the LOO procedure) and did not appear to take into account the number of variables in each model. If this is correct, this goes against the sort of penalties that traditional approaches include for models with similar fit but with more variables. In general, if one of the goals of the paper is to develop a new framework for analysis, it might be worthwhile exploring the costs and benefits of using different approaches for model evaluation.

Fourth, I understand the idea of using the term looking as a more descriptive term for vigilance. However, in many species, it might be very difficult to determine gaze direction and thus what animals are looking at. This is relatively easy for species with frontal and mobile eyes. But for birds, for instance, this is more difficult as we do not really know where they are looking plus their eyes have limited movements. The frequency and duration of looks would be difficult to calculate. Therefore, researchers have used postures (head up or down) or the frequency of head movements as proxies for vigilance.

Fifth, the authors should make it very clear that there were studying one large group with all the constraints that may apply : we do not know if the results are replicable across groups and if they would be similar in smaller groups.

Sixth, the title to me is not really representative of the main findings in this paper. Vigilance is needed to detect threats but may not always be costly. Perhaps put more emphasis on which drivers are important in this particular species with very specific constraints (one large group, one primate species with frontal eyes).

I have also some minor issues listed below by line number.

Line 311: Please explain how individuals could be distinguished in the field.

Line 321: So profile videos were acceptable but not if the animal shifted its head in the other direction, right? So information from one eye was deemed sufficient to infer the target of looking, which makes sense as the two eyes typically look in the same direction.

Line 346: Perhaps a quick justification for the 5 m rule would be in order.

Line 358: The definition of edge: I am a little surprised that an individual with 5 companions closer to the edge than themselves would still be considered at the edge. In terms of dilution of risk, this would seem just as sufficient as for those more inside. Perhaps such cases were rare. There are other definitions of what constitutes the edge with the simplest using the contour of the group.

Line 432: I think it should be made clear early on that mixed models were used with id as a random factor to account for repeated measurements of the same individuals. Also, the term behaviour in table 1 was not really properly explained in this section. As an overall comment here, is this use of Bayesian modelling really necessary? It seems to me it would be much simpler to calculate look frequency as looks per min and look duration as the percentage of time spent looking. Then linear mixed models could be used to compare models. Are we really gaining anything by using sophisticated methods that take pages to explain?

Line 476: I am not too familiar with this stacking procedure. Is there a reason why the usual comparison of models with AIC was not used? Please provide more information about what the terms weights and shared mean in Table 1.

Line 485: I did not see Table 2.

Line 242: Perhaps the lack of effect of spatial position was related to the definition used in this case, which I found to be quite generous as pointed out earlier.

Line 277 : Here the authors use the term vigilance instead of looks. I think this can be confusing.

Reviewers' comments:

Reviewer #1 (Remarks to the Author):

The manuscript investigates vigilance behaviour within a (previously proposed) broader 'looking framework' that encompasses multiple competing hypotheses (social threat, tolerance to human observers and predator detection), using chacma baboons as a model system.

It is generally well-written, identifies an interesting, topical issue that merits publication and is an impressive analysis of a focused dataset. However, I did find that the description of data collection lacked clarity in many places. For example, several terms/categories are introduced in the main section and results that are not mentioned in the Methods, making tracking how data were collected and what is being analysed a little difficult. I had one issue with the analysis of social threat and the measure calculated. I hope the comments below will be helpful in improving the paper.

Corresponding author: We thank the reviewer for their feedback, we agreed with many of their comments and have used them to improve the consistency of terminology and description of our methods.

Introduction

Line 43 Identity = identify

Corresponding author: Thank you, this has been changed.

Line 56 I think the term 'search' here is a little confusing (also line 20). As the authors' know, it wasn't specified in the Allan and Hill definition but was attributed to the Treves definition 'any visual search or scanning "directed beyond an arm's reach"'. The term search implies a goal oriented behaviour and subjectivity in deciding whether looking involves searching or not.

Corresponding author: Thank you, this has been updated to "In this approach, observers record a general visual search behaviour – looking - across a full range of scenarios, regardless of the study animals' posture (e.g., head up) or internal state (e.g., vigilant, cautious etc)." See lines 73-74.

Results

Line 72 'foraging task' does not appear in the Methods or the supplementary materials making it difficult to match with data collected. 'Specific behaviours' also does not appear in the Methods/data collection protocols although it is mentioned in the model rationale in the supplementary. Please be consistent in use of terms throughout and define these categories in your data collection section. Similarly, within-group threats in referred to the next line, is this within-group risks in Table 1 (Model 15)?

Corresponding author: Thank you, this is important to make clearer. We have now added the feeding rate/food items, specific behaviours, and within-group risk labels in the appropriate places throughout the methods (see lines 480-481 and 492-493 to the new Table 1, and the results Table 2. Within-group risks has been updated to within-group threats throughout.

Line 84-85 What is the difference between an extra-group member and a foreign individual, and if you make a distinction how did you distinguish between them?

Corresponding author: There isn't a difference in our analysis as we treated any foreign individual as representing an extra-group conspecific threat, whether they be lone individuals, individuals temporarily leaving their usual group, or an entire group. This was done as even lone individuals stimulated group-wide alarm responses. We have made this clearer now, see lines 507-512.

Line 108 Table 1. The sub-heading 'Compatibility factors (specific behaviours and feeding rate)' is a little misleading as only Model 5 is referred to as the specific behaviours model. Perhaps a different sub-title would be appropriate here so that Model 4 with 'Behaviour' can be included. Is part of the description for Model 5 missing? 'Within-group risks' is separate from the sub-section of pre-emptive risks but included in pre-emptive vigilance in Table S2, can you re-name? Observer movement is missing from the

Methods and only appears in tables. Can you please describe how it was collected?
Please define 'vis' in the Table caption and use vis consistently throughout the table.

Corresponding author: These are important points, and we thank the reviewer for taking the time to read our supplementary information so thoroughly. We have kept the Compatibility sub-heading, as both models explore our hypotheses on compatibility, but changed the part in parentheses to "feeding rate/food items and specific behaviours".

The 'pre-emptive social vigilance' label in Table S2 (now Table 1) was a mistake and has been changed to 'within-group (social) vigilance' in keeping with the terminology used in the main text.

We have updated the observer tolerance section of the methods to detail how observer movement was collected (see lines 547-553). We have also updated Table 1 and Table 2 (formerly Table 1) so that visibility is written throughout the table in keeping with the methods.

Line 126 Self-grooming, also referred to as auto grooming in your tables, please be consistent in terminology.

Corresponding author: Thank you, this has been changed in the tables to self-grooming.

Discussion

Line 185 The initial summary a little vague and it would be useful to refer to the frequency and duration of vigilance here.

Corresponding author: We are running tight to the word count and so preferred to keep this paragraph as general as possible. The rest of the discussion details the specifics of these effects on the frequency and duration of looking.

Lines 227-8 It's not clear here, or in this paragraph, if the authors are talking about frequency or duration of vigilance or both.

Corresponding author: Thank you, we have added 'increased duration of looking' to the opening sentence to make this clearer (see line 261).

Line 242 Given that male baboons have often been found to often occupy the edges of the group it would have been interesting to see interaction effects with age-sex class for

the pre-emptive models. Why wasn't this included and is this a limitation that you can discuss?

Corresponding author: We do agree that the notion of male's occupying peripheral locations has characterised baboon ecology for sometime. However, in our experience, these factors aren't true and are better described at an individual level. Across 3676 focal observations, three of the four most observed individuals at the front/periphery of the group were adult female. When considering 'peripheral' alone, the top 10 individuals were all adult female (and all lower ranking). We believe this makes sense ecologically as male baboons are often socially dominant scroungers (which I have also observed in natural and experimental scenarios across several groups in Southern Africa), and therefore will often be found monopolising high-quality food patches in the centre of the group, usually surrounded by lower-ranking individuals waiting for access (i.e., queueing). This creates a pressure for the other individuals to move on and search for new foraging patches, placing them at the periphery of the group. We therefore think the reviewer's point should extend not as an interaction with age-sex class, but operate at an individual level. These factors therefore do not bias our results as the random effect of individual identity is always in place, thus when spatial position is included as an additive fixed effect (alongside age-sex class), the model is asking whether looking is varying as a function of spatial position (with variance already partitioned towards age-sex class and individual identity), thus answering whether the study animals generally increase/decrease looking according to spatial position.

We agree that an even more nuanced question is also possible – whether individuals vary in their response to being in different spatial positions than they typically are. But such an analysis would be very data-hungry (i.e., spatial position/cohesion included as population-level terms and additionally as slopes over individual identity) and impossible to fit with the sample sizes typically observed in animal vigilance research. This is an interesting avenue for future research to consider however. We do already mention the need for random coefficients/slopes already (lines 575-581), but we have added a summary of this information to the discussion and now highlight it as a future topic for researchers to consider (see lines 294-309).

Line 246 Methodological consistencies may play a role but for data collected on the same group by some of the same researchers, this explanation is a little weak. The citation here is not published but under review – is this data on the group from the same time period? If so, what specifically about the analyses could account for these differences? Was the same definition of looking behaviour utilised? How were data collected differently. This is an important point and suggests that the findings here are not necessarily robust, unless further explanation can be offered.

Corresponding author: Good point, we were trying to be concise but accept these details are interesting and discussion is in line with our previous work. We now include some more information on our other study (see lines 283-286). This data was collected in the years leading directly upto the current study and by the same authors. Most of the individuals in the study group were the same. We used the Treves definition in that study, which partially motivated our decision to suggest the looking definition and framework (discussed in Alan & Hill, 2021). The former study also used instantaneous point sampling. We now make reference to other vigilance studies showing how different terminologies may lead to different results, including our own previous work on definition and interpretation effects.

Line 278 From what you say above, this seems contradictory with your other research under review?

Corresponding author: Yes, we have highlighted this (lines 282-283).

Line 298-9 The approach outlined, whilst all encompassing, reflects an enormous body of work from a large team of researchers. Recognition of the resources required to repeat such an approach is important here I think.

Corresponding author: All focal data (and most of the accompanying contextual data) was collected by the corresponding author, accompanied by a single additional researcher at a time (who gathered ranging data and adlib data on alarms, aggressions, and interactions with other species/groups). This is in keeping with most vigilance studies on animals and requires minimal resources. We always kept the number of observers in the field low to minimise the likelihood of disturbing observations and

creating a human-shield effect by scaring away predators (see LaBarge et al. 2022 for evidence that reducing observer numbers at this same field site can mitigate this effect).

Methods

Line 310-313 Mention here that individuals were individually identifiable. This isn't explicitly stated but is then deduced further down where a focal list is created for observations.

Corresponding author: Thank you, we have added this (lines 376-377).

Line 315 How did you arrive at 30-sec video samples?? Can you report looking distributions as the authors recommended in Allan and Hill 2017?

Corresponding author: This is a good point, we initially ran a pilot study using four minute focal observations and ran an analysis comparing sub-sampled 60- and 30-seconds focal lengths. Sub-sampling did not affect results or the distribution of bout lengths (most were less than 3 seconds). This coupled with the challenges associated with longer focal lengths (aborted samples were much higher for 4 minute and 1 minute observations than 30 seconds) motivated our decision to reduce the focal length to 30 seconds. The benefit was this enabled us to observe all the baboons across a greater variety of ecological scenarios. We have included a new section in the supplementary material (text S3) to detail all of this information.

Line 316 What field of view (around the whole? animal or face) was recorded when focal observations were taken? This seems important considering the definition (see comments below about arm's reach). Was there a protocol for video-ing that included a minimum approach distance?

Corresponding author: Thank you, this has been updated to highlight (see lines 404-411) that we tried to keep the focal animal and ~2.5 meters in view at the onset of the observation, but that the field of view often narrowed if the baboon reduced its proximity to the observer (e.g., when moving between sub-patches during foraging or during play/aggressions). Vegetation and cliffs often made it impossible for the observer to adjust position, whilst taking backwards steps when baboons approach is always

avoided as it may contribute to 'problem' animals developing (i.e., viewing observers as subordinate and therefore redirecting aggression towards them).

Line 317 'four time-periods' – did you attempt to sample equally within each time period and did you avoid sampling twice on the same day? The crossed random factors of date and ID suggest repeated sampling on the same day. If so, how many different observation days are represented in the 54-59 observations per individual? For example, are there 15 days of data for an individual with observations in each time of the four periods in one day? A better explanation of how the data were collected would help the reader understand the quantity (and reliability) of the dataset.

Corresponding author: Thank you, we have now included that there were 78 observation days. No individuals received more than 2 observations per day. Each individual generally received 14 observations per time period, however, some individuals received less (minimum 12) in certain time periods and more (maximum 18) in others. See lines 397-401.

Line 327 Whilst the number of observations per individual ranged from 54-59, this constitutes 27 minutes per individual and 29-h of data for the group. Could you explain about how this was distributed over the 14-month fieldwork period? For example, was each individual sampled twice a month?

Corresponding author: We have updated this section to be clear. The focal observations were split into two periods, June-Dec 2018 and Feb-June 2019. We attempted to sample individual equally in each year (29-32 per individual in 2018 and 25-28 in 2019). Given our sampling methodology, individuals were evenly sampled across time periods during each block. This generally resulted in ~1-3 focal observations per month in each time period. See lines 389-397.

Line 332 I find the definition 'its line of vision extended beyond its hands and the substrate.' unclear, This could be taken to mean the substrate e.g. ground is always excluded from looking behaviour or only the immediate substrate, beyond the hands or in an area around the animal. Later on at line 337-40 it is partly clarified that large objects in arm's reach (trees, rocks) were considered substrate and so excluded. This means the line of vision must extend beyond arm's reach – can you add this to your

definition? Does this also mean a tree or rock 1.5m away was included in looking? Line 335 also states that the animal 'had to divert its line of vision away from its hands'. In the definition, looking 'beyond,' could just mean looking past the hands and so the hands could be in line of sight – can you add that the line of vision had to be diverted from hands? The information at lines 337-40 needs to be incorporated into the definition as these criteria are crucial elements.

Corresponding author: The whole paragraph should be viewed as 'the definition'. As we recommend in our previous studies, researchers generally do not give enough detail on how to operationalise definitions, nor examples of its appropriate implementation. The key is that the definition should be individual specific, i.e., distance rules are meaningless as baboons have different arm lengths (other species have different eye heights, neck lengths etc). The key is about the focus of an individual's attention and whether their vision is obscured, and thus their ability to detect threats is hindered, we have amended lines 428-434 to make this clearer.

We also agree on the distinction between looking past the hands and diverting the line of vision away from the hands – we think these are maybe subtly different and now include the latter in the 'main' definition as it is how we intended the definition to be interpreted and implemented, see line 420.

We also now include reference to our previous study where we validated our definition, investigating whether an animal that was looking, not looking (but not engaged in a task), or not looking (but engaged in a task), influenced the distance at which baboons detected approaching observers. Although baboons are generally good at detecting threats regardless of behaviours, looking clearly promoted earlier detection. We think this helps contextualise the goal of the definition and now offer examples of how others may adjust it for other species, see lines 438-447.

Line 346 It would help if 'spatial cohesion' could be added here as it took me a little time to search through to find out how cohesion was calculated.

Corresponding author: Good point, updated.

Line 356 It seems the periphery of the group was determined through sightings and

audible cues – were these collected by walking around the perceived group perimeter after each sample, or collected from the observer position after the video recording? If the latter, this would seem open to bias? Additionally, as a sex bias has been noted in baboon individuals with males more likely at the front of the group e.g. van Schaik et al 2021, Fig.2 (DOI: 10.1111/eth.13233) and more likely engaged in vigilance (Matsumoto-Oda et al 2018) can you discuss what impact excluding this has on the results?

Corresponding author: Thank you, this has been updated (lines 464-470). It was assessed before, during, and at the end the observation (as this can change quite quickly), the observer then recorded the spatial position the focal animal had spent the most time during the focal observation. This was done from the observer's position when it was unambiguous (e.g., high visibility spots allow a good view of the entire group) but if visibility was an issue, then the observer(s) quickly assessed the broader area to be sure. We have addressed the sex bias/spatial position point in our response to the reviewer's earlier comment regarding line 242.

Line 360-79 Can you include an ethogram (descriptions) of your behaviours e.g. agonistic interactions span many types of behaviour in the supplementary? Similarly, movement can be collected in different ways.

Corresponding author: We have added additional examples to the aggression/play and movement categories (lines 482-485 and 487). The rest of the behaviours are operationally defined already.

Line 371 Not engaged implied no visual attention required, and in the model justification it was used as a predictor for duration of looking. Mating is included here but could refer to copulation (visual attention on the female?) or consortship (visual attention on the female and surroundings?). The rationale for including it here is not clear. Drinking behaviour is also classified as not engaged but would seem to involve head down and thus visual attention is occupied? Greater explanation in an ethogram might resolve this lack of clarity.

Corresponding author: Mating does not seem to require the individuals involved to look at one another. Animals can look beyond/away from their hands during the behaviour –

and in fact it is regularly observed that both individuals will look around, potentially because they are vulnerable to attack at this point and wish to monitor threats/competitors. The female can of course look at the ground and the male at the female's back (in contact with their hands), and therefore be 'not looking', however, this isn't constraint of mating. Similarly with drinking, the animal can look at the water or the substrate they are touching (e.g., rock they are balancing on), but it is not needed as the animal can also look around and still drink effectively. The definition has no postural requirements, but we have added a drinking example to our definition to improve this interpretation (see lines 428-432).

Line 380. After stating that events were recorded if ongoing, line 392 then says 'no event (during the day so far)' and that timing post-event was calculated– does that mean you also separately recorded all occurrences or ad lib events throughout the day? If so please add this sampling method.

Corresponding author: Thanks, we have added this (lines 511-512).

Line 402 Number of social threats reflects the number of higher ranked non-clique neighbours but this metric doesn't distinguish between e.g. a situation where there is a single higher-ranking non-clique neighbour within 5m (a value of 1), and a situation where there are three higher-ranking neighbours including two higher-ranking clique neighbours (a value of 1) but the social threat level should be much higher in the first context. A better measure might be to divide your metric by the total number of higher ranked neighbours.

Corresponding author: It is a count of higher-ranking non-clique members and thus explores the effect of increasing social threats. Total number of neighbours also included in the model to account for group cohesion.

Line 430 Insert 'how'

Corresponding author: This paragraph has been completely re-written according to other comments (see lines 547-553)

Line 431 In the Supplementary S7, there is much reference to updating (47). Is this a reference – if so it's Vehtari, A., Gelman, A., Gabry, J. & Yao, Y. loo: Efficient leave-one-out

cross-validation and WAIC for Bayesian models. R package version 2.0.0, which makes the methodology a little hard to follow. Can you also define FID at first mention before using the abbreviation.

Corresponding author: The reference list in the supplementary material is for the supplementary material only (as required by the journal). Therefore, (47) refers to our previous study on Visual Orientation Distance (VOD) and Flight Initiation Distance (FID) conducted on this group of baboon previously. Lines 549-553 to the main text to describe how VOD was measured and what it means.

Line 446 As Table S2 presents the hypotheses tested by the study I think it should be moved to the main part of the paper rather than in the supplementary (where detailed justification of the models is given).

Corresponding author: We agree and have moved this table (now Table 1) to the end of the introduction (see line 90).

Lines 459-63 This information is about creating your models and model structure so would be better placed up at line 446.

Corresponding author: Agreed, moved (see lines 581-585).

Supplementary

End of page 2, 'Each response variable also included a model (Model 5) designed..' there should be more model numbers referenced here?

Corresponding author: Each response variable was run with model 5's notation to test for hypotheses relating to specific behaviours (e.g., picking, grooming, handling etc), model 4 explored the inherent compatibility of specific foraging tasks (e.g., foraging on Acacia karoo seeds vs grass roots).

Table S2 I think model 12 is missing the word 'neighbours' at the end? Model 7 is included in the sub-section Reactionary risks in Table 1 but there is no rationale for Model 7 in the text that follows or mention of reactionary vigilance.

Corresponding author: Good spot, thanks. The row height cut 'neighbours' off.

There is some information justifying model 7 in Table 1 but we have added some more information to text S1 – this relates to social information acquisition generally. For

example, as baboon males often compete to form consortships with females, they may look around more during or immediately after seeing/hearing a copulation. Elsewhere, juveniles are also known to 'protest' copulations involving their mothers.

Corresponding author: We'd just like to reiterate our thanks to the reviewer for their detailed feedback, we think it has greatly improved the understanding of our study and its methods.

Reviewer #2 (Remarks to the Author):

In this paper, the authors investigated the drivers of looking in a large troop of baboons. What is new here is the large number of variables included in the analysis including various families of drivers related to food, predation and social environment. The discussion of the findings is clear and illuminating. The results confirmed several patterns established in other species and uncovered novel ones like the effect of observers. The scope of the analysis is really broad and unique. I would like to raise some caveats about the analysis in the spirit of transparency. They should not be viewed as negative comments.

Corresponding author: We thank the reviewer for reading our work and offering some excellent feedback which we have used to improve our manuscript.

First, looking was investigated across the day and so presumably it includes episodes during which the animals were resting, socializing or foraging. The drivers of looking might not be the same during these various episodes. For instance, when feeding, looking might be affected by the sort of food eaten and less by social factors. Social factors might play a larger role when animals are resting or grooming. Much of the literature on vigilance typically focuses on specific contexts such as resting or foraging but rarely across contexts like in this study. Can this explain some of the findings?

Corresponding author: This is a good point and something we want to bring to attention with our approach. Based on this feedback, we have made a few changes. Firstly, we now highlight in the introduction that there are a potentially endless list of nuanced vigilance

hypotheses. Most research tries to tease these apart by recording subtypes of vigilance in the field (e.g., social vigilance) and under specific scenarios (e.g., only during feeding). The drawbacks of this approach are that it is challenging to sample specific subtypes of vigilance and that doing so still leaves researchers unclear what contribution vigilance specific scenarios has to animal lives generally (lines 61-68 and 39-44).

Secondly, although we already highlight in the methods (lines 574-581) that many of these nuanced hypotheses (e.g., social vigilance during foraging) would ideally be tested using some very complex models that include numerous interaction terms at both the population-level and as random slopes over individual identity, we now include a section of the discussion devoted to this point (see lines 294-309). Unfortunately, most research lacks the sample sizes required to do so, and instead explore these hypotheses in isolation to one another (e.g., vigilance drivers during feeding).

Despite this, we believe our approach offers a route around this sampling burden. In our framework, behaviour (time spent engaged/not engaged) is included in all models, as an additive effect in reactionary and within-group threat models (as these threats are local and realised and therefore should draw visual attention regardless of behaviour) and as an interactive effect in pre-emptive and observer models (as the effects may be much more subtle). In the latter case, we found little evidence that behaviour interacted with pre-emptive risk variables (e.g., home range familiarity, leopard RSF) were good predictors of looking bout frequencies or duration. What we tend to see is that, if a baboon has an opportunity to look, then it does, regardless of external factors, promoting it's chances of acquiring useful information, including on threats. Our new paragraph in the discussion now covers all of this (see lines 294-309).

Second, the analysis focuses on different families of drivers (geometric, within-group risks, reactionary risks, etc.). As far as I could tell, model comparisons did not consider cases mixing different families of risk factors. I understand that the paper already considered 21 models, but it should be worth mentioning that other models in the future could include combinations of various types of risk factors.

Corresponding author: Agreed, we addressed this as part of our response to the previous point.

Third, again as far as I can see, model comparisons rested on predictive power alone (using the LOO procedure) and did not appear to take into account the number of variables in each model. If this is correct, this goes against the sort of penalties that traditional approaches include for models with similar fit but with more variables. In general, if one of the goals of the paper is to develop a new framework for analysis, it might be worthwhile exploring the costs and benefits of using different approaches for model evaluation.

Corresponding author: The advantage of stacking to average Bayesian predictive distributions, is that the approach does not penalise models according to their complexity, instead comparing them according to the precision in predicting 'new' observations. The Leave-One-Out Cross Validation (LOO-CV) procedure removes each observation in turn and uses the remaining data to predict this 'left-out' observation, models that consistently predict this point with the most precision garner the strongest stacking weights. The result is a framework that allows any well-fitted model to be used without imposing a fix penalty as AIC does. We have now added this detail to our methods section to highlight this and several other benefits (see lines 603-604 and 609-613). The drawbacks of our approach used to centre around the time it took to manually calculate each LOO-CV estimate for every data point, but modern computing software have practically removed this burden. As a result, we don't really see any cost to undertaking this Bayesian approach. Stacking has already been shown to outperform BIC and WAIC, the latter is analogous to AIC, but is widely considered an improvement.

Fourth, I understand the idea of using the term looking as a more descriptive term for vigilance. However, in many species, it might be very difficult to determine gaze direction and thus what animals are looking at. This is relatively easy for species with frontal and mobile eyes. But for birds, for instance, this is more difficult as we do not really know where they are looking plus their eyes have limited movements. The frequency and duration of looks would be difficult to calculate. Therefore, researchers have used

postures (head up or down) or the frequency of head movements as proxies for vigilance.

Corresponding author: We can understand this comment, but are careful to use the phrase 'line of vision' in our work, as definitions relying on terms such as gaze are typically unreliable, even in baboons – see research exploring how different vigilance definitions produce varied results in baboons (Allan and Hill, 2021). We believe the looking definition can apply to other species, including birds – in all case it likely relies on video-sampling. This was actually discussed in our previous study (Allan and Hill, 2021), but was not discussed here as we wanted to keep the definitional discussion separate to the analytical framework discussion. We accept that this may not be useful however, and now highlight some of this in our introduction (lines 69-78). We have also edited the methods (see lines 428-434 and 438-447) to make it clear that the key to implementing our definition in other species is appropriate consideration, testing, and validation of detection capabilities in different postures/behaviours (as we did with the baboons, see Allan, Bailey, and Hill 2020). For example, we know that the baboons have improved threat detection when their line of site extends beyond or away from their hands, body, substrate, or object/animal they are in contact with. In Allan and Hill (2021), we suggest that researchers test head up and head down postures with obstructions within/beyond a wing's length. For example, birds may have their head up, but will not be able to detect predators if vegetation is obscuring their line of sight. Equally, a head down bird can detect predators as well as head up birds if their visibility is unhindered. Fifth, the authors should make it very clear that there were studying one large group with all the constraints that may apply : we do not know if the results are replicable across groups and if they would be similar in smaller groups.

Corresponding author: Agreed, but we do discuss this on lines 287-289. Our framework would produce different results if indeed a smaller group was more risk sensitive. For example, the specific behaviours and feeding rate/food items models may have less weight, whilst the leopard risk or group geometry and cohesion models may have more. Sixth, the title to me is not really representative of the main findings in this paper.

Vigilance is needed to detect threats but may not always be costly. Perhaps put more

emphasis on which drivers are important in this particular species with very specific constraints (one large group, one primate species with frontal eyes).

Corresponding author: We disagree that vigilance is needed to detect threats. Vigilance is subcomponent of looking and really the internal state of being watchful for dangers or threats. But an animal looking at a tree can detect a predator if it walks near the tree, it isn't using vigilance to monitor the tree in the first place – we discuss this in previous work, but is highlighted by numerous researchers now. We doubt our findings will prove to be specific to this group of baboons, or indeed this species or order. The key is that we aren't measuring vigilance. We are measuring looking (based on line of sight). We know that birds and other mammals can detect threats in head down postures or even when their view is partially obscured, but tend to lose this ability when heavily focused or their view is mostly obstructed. Many species (including birds) can also detect threats when manipulating food items (mentioned in the discussion – lines 245-246). For these reasons, the assumption that birds require a head up posture to detect threats is clearly not unanimously correct.

The first part of our title describes our study (A framework for disentangling the behavioural and risk drivers of looking), the second part is specific to our findings on the baboons but should apply elsewhere (Why some animals may not need vigilance to avoid threats). Our results show that the baboons rarely seem to engage in pre-emptive vigilance (the subtype of vigilance most researchers are interested in), probably because many of their behaviours have compatible looking time. When combined with our past research (e.g., Allan, Bailey, and Hill 2020) showing that these baboons can detect threats quickly when looking around, but also perform well even when engaged in other tasks/behaviours, it seems likely they are under little pressure to be pre-emptively vigilant. This is a central part of our discussion and the culmination of several studies that were carefully designed to explore exactly these points. The amendments we have made to the introductions (lines 39-44 and 69-78) and methods (lines 428-434 and 438-447) should now make it clearer why the title is a good fit to our results.

I have also some minor issues listed below by line number.

Line 311: Please explain how individuals could be distinguished in the field.

Corresponding author: Initially via easily distinguishable features (e.g., scars, distinctive pelage etc) but eventually, experienced observers will identify individuals instinctively, as the corresponding author did for this study (see lines 375-377).

Line 321: So profile videos were acceptable but not if the animal shifted its head in the other direction, right? So information from one eye was deemed sufficient to infer the target of looking, which makes sense as the two eyes typically look in the same direction.

Corresponding author: Yes, this is correct, we have added 'and one eye completely' to line 403.

Line 346: Perhaps a quick justification for the 5 m rule would be in order.

Corresponding author: Thanks, this is a good point, it was derived from pilot testing. We have added this information/data to Supplementary Text S3.

Line 358: The definition of edge: I am a little surprised that an individual with 5 companions closer to the edge than themselves would still be considered at the edge. In terms of dilution of risk, this would seem just as sufficient as for those more inside. Perhaps such cases were rare. There are other definitions of what constitutes the edge with the simplest using the contour of the group.

Corresponding author: Thanks again, we have added to this paragraph to make our method clearer (see lines 464-470). As the group was 90 -100 individuals, 5 individuals represented ~ 5% of the group. So, even if they were all immediately 'outside' of the focal animal, this doesn't represent a significant buffer, especially as spacings were irregular. Although five local individuals may be 'further away' from the rest of the group, they may not be directly between the focal and the edge, or even close enough to influence dilution effects.

Line 432: I think it should be made clear early on that mixed models were used with id as a random factor to account for repeated measurements of the same individuals. Also, the term behaviour in table 1 was not really properly explained in this section. As an overall comment here, is this use of Bayesian modelling really necessary? It seems to me it would be much simpler to calculate look frequency as looks per min and look duration

as the percentage of time spent looking. Then linear mixed models could be used to compare models. Are we really gaining anything by using sophisticated methods that take pages to explain?

Corresponding author: We agree that the random effect information seems a little late, this has been moved further up, see lines 584-585.

We don't think the Bayesian approach is any more sophisticated to run with the advent of modern software. In fact, the brms package uses lme4 syntax; thus, running our models would have almost identical script whether frequentist or Bayesian. Most of the length of our methods is derived from being thorough when describing our generalized linear mixed-effect models, including the necessary diagnostic tests and model comparison procedures. The benefit of our approach is the LOO-CV and stacking procedure, which research has shown to be a vast improvement on comparing the AIC values or WAIC weights of different models (see lines 603-604 and 609-613).

Although past research has divided observations by sampling effort and analysed this number directly (e.g., frequency of looks per min, or duration of looking as a proportion of the sample), this approach is problematic as the denominator then varies across the dataset, leading to biased model estimates. Our approach is favourable (and not inherently Bayesian), whereby the response variable (duration or frequency) is offset by the sampling effort (focal duration), i.e., the models now implicitly control for different observation lengths in the posterior distribution. No other vigilance study that we are aware of account for curtailing vigilance bouts, again, this isn't Bayesian, and should be a necessary component of any analysis that has censored observations.

Line 476: I am not too familiar with this stacking procedure. Is there a reason why the usual comparison of models with AIC was not used? Please provide more information about what the terms weights and shared mean in Table 1.

Corresponding author: Thank you, we have added several lines to these paragraphs to fully explain the benefits of LOO-CV and stacking over AIC (see lines 603-604 and 609-613) and what the weights mean (see lines 615-617 and 620-621).

Line 485: I did not see Table 2.

Corresponding author: Thanks this was a typo, but we have added a new table 1 and so this reference is now correct.

Line 242: Perhaps the lack of effect of spatial position was related to the definition used in this case, which I found to be quite generous as pointed out earlier.

Corresponding author: We think this is unlikely because five individuals isn't many and usually included juveniles. In most cases, peripheral focal individuals would have been the most peripheral individual, or at least had only one or two non-infant individuals directly between themselves and the 'edge'. In cases where 5 local individuals were further away than the focal animal, none of these observations were instances of a subgroup of individuals clustered together. Usually, individuals are relatively widely spaced, so dilution effects would have been minimal. We agree that these details are missing from our previous version of the manuscript and so have updated the methods accordingly, see lines 464-470. If being isolated was a consistent driver of elevated looking patterns (inferring vigilance use), we'd expect it to have some weight in at least one stack, but none of the models including this variable had considerable weight. We are therefore very confident in this conclusion; however, we do now suggest that investigating isolated individuals explicitly could be an avenue for future research, see lines 278-280.

Line 277 : Here the authors use the term vigilance instead of looks. I think this can be confusing.

Corresponding author: Thanks, we have adjusted the wording. See lines 332-333.

REVIEWERS' COMMENTS:

Reviewer #1 (Remarks to the Author):

I'm glad the authors found comments useful and thank them for giving my feedback careful consideration. I have just three comments outstanding:

Line 56 I think the term 'search' here is a little confusing (also line 20). As the authors' know, it wasn't specified in the Allan and Hill definition but was attributed to the Treves definition 'any visual search or scanning "directed beyond an arm's reach"'. The term search implies a goal oriented behaviour and subjectivity in deciding whether looking involves searching or not.

Corresponding author: Thank you, this has been updated to "In this approach, observers record a general visual search behaviour – looking - across a full range of scenarios, regardless of the study animals' posture (e.g., head up) or internal state (e.g., vigilant, cautious etc)." See lines 73-74.

Reviewer response: Thank you, but this still retains the subjective term 'search'. Why not re-phrase this to "In this approach, observers record general looking behaviour across a full range of scenarios, regardless of the study animals' posture (e.g., head up) or internal state (e.g., vigilant, cautious etc)." ?

Line 298-9 The approach outlined, whilst all encompassing, reflects an enormous body of work from a large team of researchers. Recognition of the resources required to repeat such an approach is important here I think.

Corresponding author: All focal data (and most of the accompanying contextual data) was collected by the corresponding author, accompanied by a single additional researcher at a time (who gathered ranging data and adlib data on alarms, aggressions, and interactions with other species/groups). This is in keeping with most vigilance studies on animals and requires minimal resources. We always kept the number of observers in the field low to minimise the likelihood of disturbing observations and creating a human-shield effect by scaring away predators (see LaBarge et al. 2022 for evidence that reducing observer numbers at this same field site can mitigate this effect).

Reviewer response: Thank you, I was also thinking of the work of reviewing and transcribing of data from video footage which can be very time-consuming. Perhaps the first author was able to complete this during non-follow days? If so, then no problem. Thinking about accompanying researchers, there was no mention of training and interobserver reliability scores anywhere – could you briefly mention interobserver reliability results or training to a criterion?

Line 402 Number of social threats reflects the number of higher ranked non-clique neighbours but this metric doesn't distinguish between e.g. a situation where there is a single higher-ranking non-clique neighbour within 5m (a value of 1), and a situation where there are three higher-ranking neighbours including two higher-ranking clique neighbours (a value of 1) but the social threat level should be much higher in the first context. A better measure might be to divide your metric by the total number of higher ranked neighbours.

Corresponding author: It is a count of higher-ranking non-clique members and thus explores the effect of increasing social threats. Total number of neighbours also included in the model to account for group cohesion.

Reviewer response: Thank you, I understood this but my point was more about the relative risk of attack? The total number of neighbours variable does not consider how many of the neighbours were higher ranking. It would be fine to acknowledge this limitation in the Discussion.

Reviewer #2 (Remarks to the Author):

Thank you for considering my comments. I am satisfied with the changes made and have no further comments.

REVIEWERS' COMMENTS:

Reviewer #1 (Remarks to the Author):

I'm glad the authors found comments useful and thank them for giving my feedback careful consideration. I have just three comments outstanding:

Line 56 I think the term 'search' here is a little confusing (also line 20). As the authors' know, it wasn't specified in the Allan and Hill definition but was attributed to the Treves definition 'any visual search

or scanning "directed beyond an arm's reach". The term search implies a goal oriented behaviour and subjectivity in deciding whether looking involves searching or not.

Corresponding author: Thank you, this has been updated to "In this approach, observers record a general visual search behaviour – looking - across a full range of scenarios, regardless of the study animals' posture (e.g., head up) or internal state (e.g., vigilant, cautious etc)." See lines 73-74.

Reviewer response: Thank you, but this still retains the subjective term 'search'. Why not re-phrase this to "In this approach, observers record general looking behaviour across a full range of scenarios, regardless of the study animals' posture (e.g., head up) or internal state (e.g., vigilant, cautious etc)." ?

Corresponding author: We thank the reviewer again for their very thorough feedback. We agree with this point and have updated this sentence to: "In this approach, observers record all looking behaviours across a full range of scenarios, regardless of the study animals' posture (e.g., head up) or internal state (e.g., vigilant, cautious etc)." See lines: 72-73.

Line 298-9 The approach outlined, whilst all encompassing, reflects an enormous body of work from a large team of researchers. Recognition of the resources required to repeat such an approach is important here I think.

Corresponding author: All focal data (and most of the accompanying contextual data) was collected by the corresponding author, accompanied by a single additional researcher at a time (who gathered ranging data and adlib data on alarms, aggressions, and interactions with other species/groups). This is in keeping with most vigilance studies on animals and requires minimal resources. We always kept the number of observers in the field low to minimise the likelihood of disturbing observations and creating a human-shield effect by scaring away predators (see LaBarge et al. 2022 for evidence that reducing observer numbers at this same field site can mitigate this effect).

Reviewer response: Thank you, I was also thinking of the work of reviewing and transcribing of data from video footage which can be very time-consuming. Perhaps the first author was able to complete this during non-follow days? If so, then no problem.

Thinking about accompanying researchers, there was no mention of training and interobserver reliability scores anywhere – could you briefly mention interobserver reliability results or training to a criterion?

Corresponding author: Thank you, we have added information in several places to make it clearer that the corresponding author solely conducted the focal observations and extraction of behavioural data from the videos, including the times when videos were coded (see lines: 366-367, 388, 391, 402-404, 450, and 460). The other researchers collected other data concerning group movements, behaviours, and interactions with other groups and species, all of which was used to test various hypotheses (see lines: 452, 497-498, and Supplementary Text S7). This approach meant that we did not run the risk of interpretation effects occurring – different definitional interpretations leading to different results amongst observers – as we showed in previous work that even when inter-rater reliability scores are very high, results can still vary across observers. We now highlight this information on lines: 402-404.

Line 402 Number of social threats reflects the number of higher ranked non-clique neighbours but this metric doesn't distinguish between e.g. a situation where there is a single higher-ranking non-clique neighbour within 5m (a value of 1), and a situation where there are three higher-ranking neighbours including two higher-ranking clique neighbours (a value of 1) but the social threat level should be much higher in the first context. A better measure might be to divide your metric by the total number of higher ranked neighbours.

Corresponding author: It is a count of higher-ranking non-clique members and thus explores the effect of increasing social threats. Total number of neighbours also included in the model to account for group cohesion.

Reviewer response: Thank you, I understood this but my point was more about the relative risk of attack? The total number of neighbours variable does not consider how many of the neighbours were higher ranking. It would be fine to acknowledge this limitation in the Discussion.

Corresponding author: Thank you, we disagree that it is a limitation in our case as intra-clique aggressions were rare; as a result, the count of social threats is a more true reflection of the number of higher ranking neighbours that are likely to be considered threatening. We agree though that this may not be the case in other groups/systems, and so we have added two sentences to highlight this point and suggest ways to test it and other similar hypotheses in future research: "As intra-clique aggressions were low in our study group, we did not explore whether the number of high-ranking neighbours (including clique members) had an effect on looking patterns, but this may be appropriate in other groups and systems. It would also be interesting to explore how looking patterns are affected by the presence/number of social threats interacted with the number of neighbouring clique members as this would identify whether affiliates can

buffer social threat perception." See lines: 245-251.

Reviewer #2 (Remarks to the Author):

Thank you for considering my comments. I am satisfied with the changes made and have no further comments.

Corresponding author: We thank the reviewer again for their useful comments that improved our manuscript.